# The tyrosine kinase KDR is essential for the survival of HTLV-1-infected T cells by stabilizing the Tax oncoprotein

Suchitra Mohanty[1], Sujit Suklabaidya[1], Alfonso Lavorgna[2,5], Takaharu Ueno [3], Jun-ichi Fujisawa[3], Nyater Ngouth[4], Steven Jacobson[4] & Edward W. Harhaj [1] ✉

Human T-cell leukemia virus type 1 (HTLV-1) infection is linked to the development of adult T-cell leukemia/lymphoma (ATLL) and the neuroinflammatory disease, HTLV-1-associated myelopathy/tropical spastic paraparesis (HAM/TSP). The HTLV-1 Tax oncoprotein regulates viral gene expression and persistently activates NF-κB to maintain the viability of HTLV-1-infected T cells. Here, we utilize a kinome-wide shRNA screen to identify the tyrosine kinase KDR as an essential survival factor of HTLV-1-transformed cells. Inhibition of KDR specifically induces apoptosis of Tax expressing HTLV-1-transformed cell lines and CD4 + T cells from HAM/TSP patients. Furthermore, inhibition of KDR triggers the autophagic degradation of Tax resulting in impaired NF-κB activation and diminished viral transmission in co-culture assays. Tax induces the expression of KDR, forms a complex with KDR, and is phosphorylated by KDR. These findings suggest that Tax stability is dependent on KDR activity which could be exploited as a strategy to target Tax in HTLV-1-associated diseases.

Human T-cell leukemia virus type 1 (HTLV-1) is a retrovirus that primarily infects CD4 + T cells in vivo. It is estimated there are 10-20 million people worldwide infected with HTLV-1, mainly in endemic regions in southern Japan, sub-Saharan Africa, South America, the Caribbean, and the Middle East[1,2]. Recent studies have revealed extraordinarily high levels of HTLV-1 infection (>40% of all individuals) in indigenous communities in Central Australia[2]. HTLV-1 is linked to the genesis of adult T-cell leukemia/lymphoma (ATLL) in ~5% of infected individuals after a 40–60-year latent period[3]. HTLV-1 infection is also associated with the neuroinflammatory disease, HTLV-1-associated myelopathy/tropical spastic paraparesis (HAM/TSP)[4]. There are four major subtypes of ATLL ranging from more indolent, slowly progressive forms (smoldering, chronic) to highly aggressive and fatal diseases (lymphoma and acute)[5]. The standard treatment options for ATLL include conventional chemotherapy, a combination of antiviral zidovudine (AZT) and interferon-alpha (IFN), and allogeneic hematopoietic stem cell transplantation (HSCT)[6]. However, chemotherapy is

largely ineffective, especially for patients with acute ATLL[6], and there is an urgent need for targeted therapies for ATLL and HAM/TSP.

The HTLV-1 pX-encoded trans-activator protein Tax exerts oncogenic functions by manipulation and exploitation of the cell cycle and host cellular signaling pathways for the expansion and persistence of infected clones[7]. Although Tax is downregulated in ~60% of ATLL, a recent study has demonstrated that primary ATLL cells have low levels of Tax mRNA and protein, but nevertheless is required for NF-κB activation and cell survival[8]. Tax is also essential for viral gene expression by recruiting host transcription factors cyclic AMP (cAMP)-response element-binding protein (CREB), the coactivators CREB binding protein (CBP), and p300 to the 5′-LTR for transactivation of the viral promoter[9]. In addition, Tax establishes an autocrine loop involving growth factors interleukin-2 (IL-2), IL-15, and their respective receptors to enhance cell proliferation by activating JAK-STAT signaling[10–12].

HTLV-1 hijacks the host machinery to induce post-translational modifications (PTMs) of viral and cellular proteins to enhance viral

[1]Department of Microbiology and Immunology, Penn State College School of Medicine, Hershey, PA, USA. [2]Department of Oncology, Sidney Kimmel Comprehensive Cancer Center, Johns Hopkins School of Medicine, Baltimore, MD, USA. [3]Department of Microbiology, Kansai Medical University, Osaka, Japan. [4]Viral Immunology Section, National Institute of Neurological Disorders and Stroke, National Institutes of Health, Bethesda, MD, USA. [5]Present address: Millipore-Sigma, Rockville, MD, USA. ✉e-mail: ewh110@psu.edu

infection and persistence. Tax undergoes multiple PTMs, including ubiquitination, SUMOylation, acetylation, and phosphorylation, to regulate its interaction with host proteins and its trafficking to different cellular compartments[13]. Tax ubiquitination is mainly lysine 63 (K63)-linked and has non-proteolytic functions because the binding of ubiquitinated Tax to proteasomes is resistant to proteasomal degradation[14,15]. Furthermore, the interaction of Tax with heat shock protein 90 (HSP90) confers resistance to proteasomal degradation mediated by K48-linked polyubiquitination in the nuclear matrix[16]. Tax also enhances HTLV-1 replication by deregulating autophagy and promoting the accumulation of autophagosomes[17,18]. These pleotropic functions of Tax indicate that it may serve as an ideal therapeutic target for the development of HTLV-1 antiviral drugs or targeted approaches for ATLL and/or HAM/TSP.

Vascular endothelial cell growth factors (VEGFs) regulate angiogenesis by binding to high-affinity receptor tyrosine kinases (RTKs), including VEGF receptors 1–3. VEGF binding to kinase insert domain receptor (KDR; also known as VEGFR2) triggers angiogenic signaling pathways, including PLCγ−ERK1/2, PI3K−AKT−mTOR, Src, and small GTPases, which regulate endothelial cell survival, migration, polarization, and vascular barrier function, as well as vasomotion during vascular development[19,20]. KDR has a tyrosine kinase insert between its two intracellular kinase domains. The Golgi apparatus serves as a hub for inactivated KDR tagged with SUMO[21]. Upon stimulation, KDR traffics to the plasma membrane, where a subset is ubiquitinated and degraded in lysosomes, whereas the remainder is recycled back to the membrane for another round of activation[22]. KDR was thought to be exclusively expressed in adult endothelial cells; however, KDR expression has been observed in multipotent hematopoietic stem cells and certain leukemias, as these cells share hemangioblasts, which are common precursors of endothelial cells[23]. KDR is functional in leukemic cells and has been implicated in increased proliferation, MMP activation, and trans-basement membrane migration[24]. Mutations in KDR have been identified in North American ATLL patients[25]; however, the expression and functional significance of KDR in HTLV-1-infected T cells remain unknown.

In this study, we used an unbiased approach to identify kinases required for the survival of HTLV-1 transformed T cells. This approach identified KDR as a critical regulator of the survival of HTLV-1 transformed T cells by controlling the stability of Tax. Furthermore, we found that a KDR inhibitor could induce Tax degradation in HAM/TSP PBMCs and selectively deplete CD4 + CD25+ cells derived from HAM/TSP patients but not uninfected controls. Together, our findings provide a strong rationale for the investigation of KDR inhibitors as a potential therapeutic in the clinic to target Tax in HAM/TSP and ATLL patients.

## Results
### KDR is essential for the survival of HTLV-1-transformed cells
To identify host proteins required for the survival of HTLV-1-transformed T cells, we conducted a kinome-wide lentiviral-based shRNA screen using the MISSION® LentiExpress™ Human Kinases with ~3200 lentiviruses targeting ~501 human kinase genes with ~6 constructs per gene. The screen was performed with a Tax+ HTLV-1-transformed T-cell line, MT-2. After lentiviral transduction of MT-2 cells and selection with puromycin, proliferation and viability of the cells were determined by the CellTiter-Glo™ luminescent cell viability assay which quantifies ATP as a measure of metabolically active cells. $Z$ scores were calculated for each shRNA (3200 total) in the library, and a cutoff of >1.5 or <1.5 of mean $Z$ scores for each gene was established (Supplementary Data 1 and 2). Only one positive regulator (KDR) and eight negative regulators (TRIB2, PRKCD, NEK5, TBCK, PAK7, RPSKKL1, STK38, and SCYL1) were identified using these criteria for mean $Z$ scores (Fig. 1A–C).

To validate the role of KDR in the survival of HTLV-1-transformed cells, we treated MT-2 cells with different concentrations of SU 1498, a specific small-molecule inhibitor targeting the active ATP-binding pocket of KDR. We examined the effect of KDR inhibition by SU 1498 on the viability and proliferation of MT-2 cells using CellTiter-Glo™. We quantified the metabolically active cells at 24 h to measure the proliferation of SU 1498-treated MT-2 cells. As expected, the proliferation of MT-2 cells was significantly impaired by increasing concentrations of SU 1498 (Supplementary Fig. 1A). For further confirmation, we performed an Annexin V/PI assay of MT-2, C8166, and HUT-102 cells treated with SU 1498 in a concentration-dependent manner for 24 h (Fig. 1D and Supplementary Fig. 1B). We detected an increase in both pre-apoptotic and post-apoptotic events caused by KDR inhibition, indicating the induction of apoptosis in HTLV-1-transformed cells (Fig. 1D). Since mitochondria play a vital role in the induction of apoptosis, and loss of mitochondrial potential (ΔΨm) triggers apoptosis, we next examined the effects of KDR inhibition by SU 1498 on ΔΨm in MT-2 cells. We measured the change in ΔΨm using a confocal-based immunofluorescence assay with MitoTracker™ Deep Red, a cationic fluorescent dye that accumulates inside the mitochondrial matrix with intact ΔΨm, and a decrease in fluorescence of the stained cells indicated by loss of ΔΨm. Confocal microscopy analysis of MT-2 cells subjected to different doses of SU 1498 treatment revealed a gradual decrease in the fluorescence intensity of MitoTracker™ Deep Red (Fig. 1E), indicative of depolarized mitochondria. Thus, KDR inhibition by SU 1498 triggers mitochondrial-dependent apoptosis of HTLV-1 transformed T cells. We also detected a significant increase in caspase 3/7 activation upon KDR inhibition as revealed by flow cytometry-based FLICA (Fluorescent-Labeled Inhibitors of Caspases) caspase assays (Fig. 1F). Moreover, SU 1498 treatment of MT-2 and C8166 cells yielded cleaved forms of PARP-1 (Fig. 1G), confirming caspase-dependent apoptotic cell death upon KDR inhibition. Consistent with these results, cell cycle analysis of MT-2 and C8166 cells treated with SU 1498 revealed G1/M arrest accompanied by a significant decrease of cells in S phase suggesting impaired cell proliferation (Fig. 1H). As a negative control, Jurkat T cells treated with SU 1498 did not undergo apoptosis as revealed by the lack of cleavage of PARP-1 (Fig. 1I). Collectively, these data suggest that KDR inhibition induces apoptotic cell death selectively in HTLV-1-transformed T cells.

### KDR inhibition elicits Tax degradation
To determine if KDR inhibition had any effect on Tax expression, we treated MT-2, C8166, and HUT-102 cells with SU 1498 and performed immunoblotting experiments. Interestingly, SU 1498 treatment triggered Tax degradation, suggesting a potential role of KDR in regulating Tax expression/stability (Fig. 2A). MT-2 cells were also treated with a different small-molecule KDR inhibitor, SKLB 1002, which significantly decreased cell proliferation (Fig. 2B). SKLB 1002 also induced Tax degradation and cleavage of PARP-1 in MT-2 and C8166 cells (Fig. 2C). We also tested the U.S. Food and Drug Administration (FDA)-approved drug Cabozantinib (Cabometyx), a small-molecule dual tyrosine kinase inhibitor of MET and KDR[26]. Consistently, cabozantinib treatment reduced the proliferation of MT-2 and C8166 cells (Supplementary Fig. 2A, B), and induced apoptosis and Tax degradation in a concentration-dependent manner (Fig. 2D). VEGFR signaling activates extracellular signal-regulated kinase (ERK)[27], a member of the mitogen-activated protein (MAP) kinase family. However, treatment of MT-2 and C8166 cells with an ERK inhibitor had no effect on Tax degradation or cell survival despite efficient ERK inhibition (Supplementary Fig. 2C). This result suggests that KDR, but not its downstream signaling, is important for the stabilization of Tax.

To circumvent potential off-target effects of KDR small-molecule inhibitors, we knocked down KDR expression using a pool of siRNAs specific for KDR. Knockdown of KDR triggered Tax degradation in

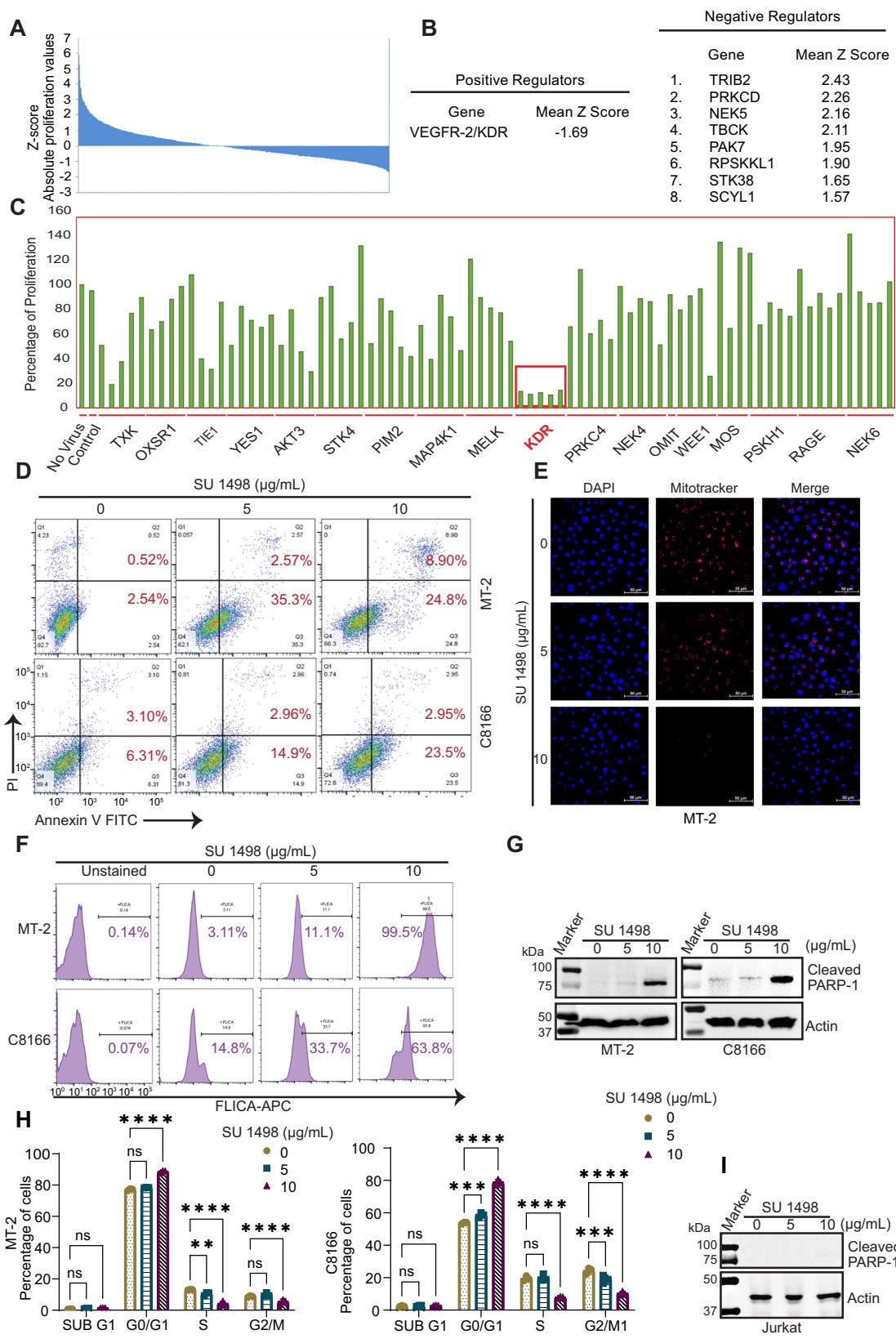

C8166 cells as examined by immunofluorescence and immunoblotting experiments (Fig. 2E, F). A lentiviral-encoded KDR shRNA (but not scrambled control shRNA) also induced Tax degradation and apoptosis in MT-2 cells (Fig. 2G). To determine if KDR inhibition could trigger Tax degradation in the absence of other viral proteins, we transiently transfected Tax in 293T cells and treated with SU 1498.

Indeed, Tax was degraded by SU 1498 in the absence of other viral proteins (Fig. 2H). Furthermore, Tax was also degraded by SU 1498 treatment in Jurkat cells expressing tetracycline-inducible Tax (Fig. 2I). Together, these results indicate that KDR inhibition induces Tax degradation, supporting a role for KDR in the stabilization of Tax.

**Fig. 1 | KDR is essential for the survival of HTLV-1-transformed cells. A** Waterfall plot representing *Z* scores for each shRNA in the kinome-wide shRNA screen for identification of survival factors in MT-2 cells. **B** Genes from the shRNA screen are listed with a mean *Z*-score of either <−1.5 (positive regulator) or >1.5 (negative regulator). **C** MT-2 cell viability in the 96-well plate from the kinome-wide shRNA screen containing the five KDR shRNAs (boxed area). **D** HTLV-1 transformed cells were treated with SU 1498 in a concentration-dependent manner for 24 h and stained with Annexin V-FITC and propidium iodide for acquisition by flow cytometry. Representative pseudocolor plots indicate the gating strategy for quantifying the percentage of each single-stained and dual-stained cells and highlighting the percentage of pre-apoptotic and post-apoptotic cell death in response to SU 1498 treatment. **E** Immunofluorescence was performed using HTLV-1 transformed cells treated with SU 1498 for 24 h and stained with MitoTracker Deep Red. The experiment is representative of two independent experiments with similar results. **F** FLICA caspase assays were performed using MT-2 and C8166 cells treated with SU 1498 at the indicated concentrations for 24 h. The histograms represent the percentage of active caspase 3/7 in treated cells. **G** Immunoblotting was performed with the indicated antibodies using lysates from MT-2 and C8166 cells treated with the indicated concentrations of SU 1498 for 24 h. The experiment is representative of three independent experiments with similar results. **H** Graphical representation of cell cycle analysis of HTLV-1 transformed cells treated with the indicated concentrations of SU 1498 for 24 h and the proportions of cells at different phases of the cell cycle (sub G1, G1/M, S, G2/M). The results are expressed as the mean ± SD of three independent experiments. ****$P < 0.0001$; ***$P < 0.001$; **$P < 0.01$; *$P < 0.05$. Two-way ANOVA with Dunnett's multiple comparisons test. *P* values from left to right: $P = 0.7880$, $P = 0.4538$, $P = 0.1483$, $P < 0.0001$, $P = 0.0010$, $P < 0.0001$, $P = 0.2411$, $P < 0.0001$, $P = 0.9872$, $P = 0.8933$, $P = 0.0005$, $P < 0.0001$, $P = 0.9264$, $P < 0.0001$, $P = 0.0002$, $P < 0.0001$, ns not significant. **I** Immunoblotting was performed with the indicated antibodies using lysates from Jurkat cells treated with the indicated concentrations of SU 1498 for 24 h. The experiment was independently repeated three times with similar results. Representative data from one experiment is shown. Source data are provided as a Source data file.

## KDR inhibition induces autophagic/lysosomal degradation of Tax

To elucidate the mechanism of Tax degradation upon KDR inhibition, we treated HTLV-1-transformed cells with Bafilomycin A1, a macrolide inhibitor of lysosomal acidification and maturation of autophagic vacuoles, or MG-132, a proteasome inhibitor in combination with SU 1498. Interestingly, Bafilomycin A1 treatment prevented the degradation of Tax, while MG-132 treatment had no effect on Tax degradation in C8166 (Fig. 3A), and MT-2 cells (Fig. 3B) upon KDR inhibition. To further substantiate these results, confocal microscopy was conducted using C8166 and MT-2 cells treated with SU 1498 and leupeptin, a protease inhibitor that promotes the accumulation of autophagolysosomes. Cells were stained with antibodies specific for Tax and the lysosomal marker LAMP2. There was a significant accumulation of Tax in lysosomes, reflected by the colocalization of Tax and LAMP2 upon KDR inhibition in MT-2 and C8166 cells (Fig. 3C, E and Supplementary Fig. 3A). However, HTLV-1-transformed cell lines treated with Bafilomycin A1 together with SU 1498 and leupeptin exhibited a significant decrease in the accumulation of Tax in lysosomes (Fig. 3D, F and Supplementary Fig. 3B). Collectively, these data suggest that KDR inhibition triggers the autophagic/lysosomal degradation of Tax.

## KDR inhibition impedes HTLV-1 replication and transmission

To investigate the effect of KDR inhibition on HTLV-1 replication, we treated MT-2, HUT-102, and C8166 cells with SU 1498 and examined the expression of HTLV-1 p19, a major core viral protein encoded by the *Gag* gene. KDR inhibition by SU 1498 triggered a dose-dependent decrease in p19 Gag expression in MT-2 and HUT-102 cells, whereas p19 Gag was not detected in C8166 cells, since these cells have defective proviruses and lack HTLV-1 replication (Fig. 4A). Similar results were also obtained by shRNA-mediated knockdown of KDR in MT-2 and HUT-102 cells (Fig. 4B). To examine the effect of KDR inhibition on HTLV-1 transmission, we used JET cells, a variant of Jurkat cells carrying a tdTomato red fluorescent protein (RFP) under the control of a Tax responsive element[28,29]. First, we confirmed that KDR inhibition did not exhibit cytotoxicity in JET cells, in agreement with our earlier experiments in Jurkat T cells (Fig. 4C). Next, we co-cultured JET cells with SU 1498-treated MT-2 cells to evaluate the effect of KDR inhibition on HTLV-1 transmission (Fig. 4D). JET cells infected with HTLV-1 from MT-2 cells de novo should express Tax and the RFP marker, and thus enable the quantification of RFP expression in real time as a marker of newly infected cells using Incucyte S3 live-cell imaging. KDR inhibition with SU 1498 impaired HTLV-1 transmission as reflected by significantly reduced RFP expression in JET cells (Fig. 4E, F). Collectively, these results suggest that KDR inhibition of infected cells blocks HTLV-1 replication and transmission.

## KDR inhibition does not affect cell death or autophagy in Tax-ATLL cell lines

The HTLV-1-transformed cell lines C8166, MT-2, and HUT-102 have high levels of Tax expression; however, studies have shown that Tax expression is silenced in approximately 60% of ATLL[30]. Thus, we wondered whether KDR inhibition would have any effect on Tax-ATLL cell lines, such as MT-1 and TL-OM1. TL-OM1 cells were treated with different concentrations of SU 1498, and the cleaved form of PARP-1 was assessed by immunoblotting; however, KDR inhibition did not induce PARP-1 cleavage in TL-OM1 cells (Fig. 5A). We next examined cell death in MT-1 cells using a FLICA caspase 3/7 assay and did not detect any caspase 3/7 activation upon KDR inhibition (Fig. 5B). Furthermore, TL-OM1 and MT-1 cells treated with various concentrations of SU 1498 exhibited no significant induction of pre-apoptotic or post-apoptotic cells as measured by Annexin V/PI staining (Fig. 5C).

Tax has previously been shown to induce autophagy by enhancing the accumulation of LC3+ autophagosomes but inhibiting the maturation of autophagolysosomes[17,18]. Given that Tax undergoes autophagic degradation upon KDR inhibition, we considered the possibility that KDR inhibition increases autophagic flux. Thus, we investigated autophagy induction by flow cytometry using monodansylcadaverine (MDC), a compound that auto-fluoresces and specifically integrates into the lipid membranes of late autophagosomes and autophagolysosomes[31,32], in Tax+ (MT-2 and HUT-102) and Tax- (MT-1 and TL-OM1) HTLV-1-infected cell lines upon KDR inhibition. The inhibition of KDR promoted the late steps of autophagy in Tax+ cells, but not in Tax- ATLL cell lines, in a concentration-dependent manner (Fig. 5D). These results indicate that KDR does not directly regulate autophagy, but Tax degradation triggered by KDR inhibition enhanced the late steps of autophagy in Tax+ HTLV-1-transformed cells. Collectively, these results demonstrate that the inhibition of KDR does not impact cell death or autophagy in Tax- ATLL cells.

## Tax induces KDR expression

To determine if Tax regulated KDR expression, we examined mRNA and protein levels of KDR in Tax+ cells (C8166, MT-2, HUT-102, and SLB-1), uninfected control leukemic cell lines (Jurkat and HL-60), and Tax- ATLL cell lines (MT-1, ATL-2S, and TL-OM1). There was increased KDR mRNA and protein expression in Tax+ HTLV-1-transformed cell lines compared to Tax- ATLL cells (Fig. 5E–G and Supplementary Fig. 4). To determine if KDR was activated in Tax-expressing cells, confocal microscopy experiments were performed using a phospho-specific KDR antibody. Indeed, there was elevated phosphorylated KDR in Tax+ MT-2 and C8166 cell lines compared to TL-OM1 cells which lack Tax expression (Fig. 5H). To determine if Tax directly regulated KDR expression, we transfected 293T cells with a Tax plasmid, and examined KDR expression by confocal microscopy. KDR expression increased in the presence of Tax suggesting that Tax

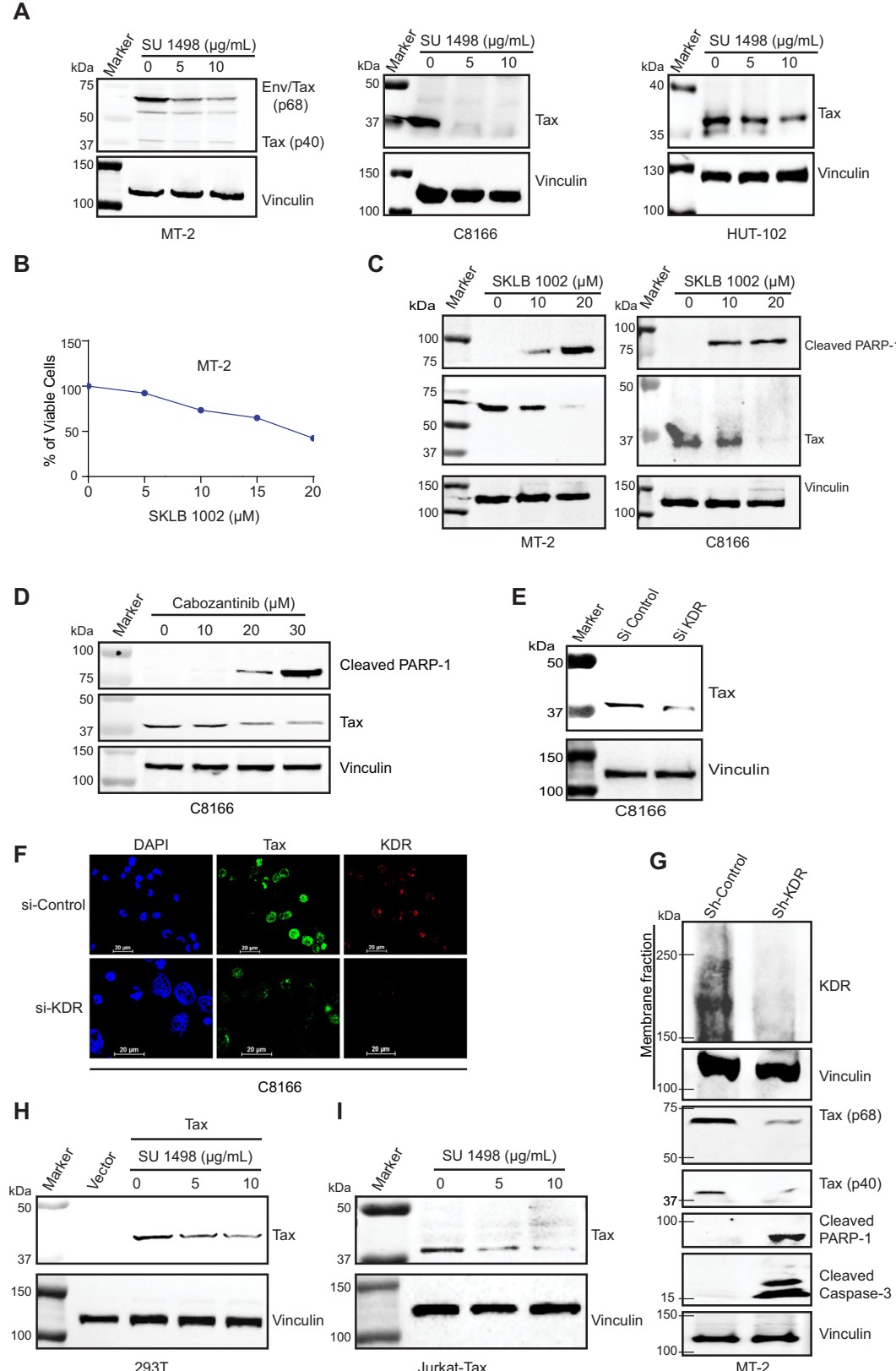

induced KDR expression (Fig. 5I). Consistent with these observations, qRT-PCR and western blotting experiments revealed upregulated KDR mRNA and protein expression in Tax-transfected 293T cells (Fig. 5J, K and Supplementary Fig. 4). Taken together, these results suggest that Tax induces the expression of KDR to prevent Tax degradation and support the survival of HTLV-1-infected T cells.

## Tax colocalizes with and interacts with KDR at the Golgi apparatus

Tax appeared to colocalize with KDR in our confocal microscopy experiments (Fig. 5I); therefore, we next sought to determine if Tax interacted with KDR to exploit its function. To this end, we performed a co-immunoprecipitation (co-IP) assay with lysates from 293T cells

**Fig. 2 | KDR inhibition induces Tax degradation. A** Immunoblotting was performed with the indicated antibodies using lysates from MT-2, C8166, and HUT-102 cells treated with different concentrations of SU 1498 for 24 h. The experiment is representative of three independent experiments with similar results. **B** Cell viability assay was performed in MT-2 cells treated with different concentrations of SKLB 1002 for 24 h. **C** Immunoblotting was performed with the indicated antibodies using lysates from MT-2 and C8166 cells treated with SKLB 1002 for 24 h. **D** Immunoblotting was performed with the indicated antibodies using lysates from C8166 cells treated with Cabozantinib for 24 h. **E** Immunoblotting was performed with the indicated antibodies using lysates from C8166 cells expressing control or KDR siRNAs. **F** Immunofluorescence confocal microscopy was performed with the indicated antibodies using C8166 cells expressing control scrambled siRNA or KDR siRNA. **G** Immunoblotting was performed with the indicated antibodies using membrane fractions and whole-cell lysates from MT-2 cells expressing control shRNA or KDR shRNA. **H** Immunoblotting was performed with the indicated antibodies using lysates from 293T cells transfected with vector control or Tax, and then treated with SU 1498. **I** Immunoblotting was performed with the indicated antibodies using lysates from Jurkat Tax Tet-On cells treated with Dox for 24 h followed by SU 1498 for 24 h. The experiments in (**C**–**I**) are representative of two independent experiments with similar results. Source data are provided as a Source data file.

transfected with Tax and KDR plasmids. Tax was detected in anti-KDR immunoprecipitates when both proteins were overexpressed, and the reciprocal co-IP also confirmed the KDR and Tax interaction (Fig. 6A). We next performed confocal microscopy to determine if Tax and KDR colocalize and to identify the cellular location of the Tax and KDR interaction since Tax shuttles between the nucleus and cytoplasm and is distributed in multiple compartments of the cell[33]. KDR has been shown to traffic from the Golgi apparatus to the plasma membrane upon VEGF stimulation in endothelial cells[34]. Interestingly, Tax also interacts with multiple proteins in the vicinity of the Golgi apparatus to activate signaling pathways (i.e., NF-κB) that promote the proliferation of HTLV-1-infected cells[35]. We previously found that Tax relocalizes NEMO and the IKK complex to the cis-Golgi using the GM-130 marker[36]. We therefore examined a potential co-localization of Tax and KDR at the cis-Golgi by staining with GM-130. Indeed, Tax-transfected 293T cells showed strong co-localization of Tax and KDR, predominantly in the vicinity of the cis-Golgi (Fig. 6B). To confirm endogenous interactions and co-localization between Tax and KDR, we conducted immunostaining and confocal microscopy in C8166 and HUT-102 cells. As expected, co-localization of Tax and KDR was observed at the Golgi (Fig. 6C and Supplementary Fig. 5). We also plotted co-localization intensity profiles and determined the Pearson's coefficient between Tax and KDR, which further supported Tax and KDR co-localization (Fig. 6D). In addition, we performed an in situ proximity ligation assay (PLA), which can detect proteins in close proximity (within 40 nm of one another). PLA specks were observed in C8166 and HUT-102 cells in the presence of both Tax and KDR antibodies (Fig. 6E, F). To further corroborate the findings of immunostaining and PLA assays, we performed co-IPs using membrane fractions from C8166 and MT-2 cells since KDR is mainly associated with membranes. Endogenous KDR was detected in anti-Tax immunoprecipitates (Fig. 6G), and the reciprocal co-IP further validated the Tax-KDR interaction (Fig. 6H). These collective data confirm an interaction between Tax and KDR under both overexpression and endogenous conditions.

## KDR inhibition impairs NF-κB signaling in HTLV-1-transformed cells

Since Tax persistently activates NF-κB signaling in HTLV-1-infected cells to drive clonal proliferation and survival of virally-infected T-cell clones[37], we next examined whether Tax degradation through KDR inhibition suppressed NF-κB signaling. C8166 and MT-2 cells were treated with SU 1498, and NF-κB activation was examined by immunoblotting. Inhibition of KDR suppressed the activation of IKK and phosphorylation of IκBα, leading to increased expression of total IκBα due to its impaired phosphorylation (Fig. 7A). Similar results were obtained by shRNA-mediated knockdown of KDR in MT-2 and C8166 cells (Supplementary Fig. 6A). To determine if the impaired NF-κB activation was caused by the accumulation of Tax in lysosomes, MT-2 cells were treated with SU 1498 and leupeptin to prevent the degradation of Tax (Supplementary Fig. 6B). Immunoblotting experiments revealed a decrease in phosphorylation of IKKα/β upon treatment with SU 1498 and leupeptin (Supplementary Fig. 6C). Therefore, although

SU 1498-induced Tax degradation was prevented by leupeptin treatment, Tax relocalization to lysosomes impaired NF-κB activation. Moreover, accumulation of Tax in lysosomes induced G1/M arrest with a concurrent decrease in S phase leading to impaired cell proliferation (Supplementary Fig. 6D). To further understand the underlying mechanism responsible for the impaired activation of NF-κB signaling, we examined the effect of KDR inhibition on the Tax-NEMO complex, which is crucial for NF-κB activation[38]. Tax-NEMO interactions in MT-2 cells treated with SU 1498 and leupeptin were examined by co-IP assays. There was a decrease in Tax detected in anti-NEMO immunoprecipitates (Fig. 7B), indicating dissociation of the Tax-NEMO complex and selective delivery of Tax to lysosomes upon KDR inhibition. These results point to a critical role of KDR in maintaining the Tax-NEMO complex for persistent NF-κB activation in HTLV-1-transformed cells.

## Phosphoproteomics identifies tyrosine phosphorylated proteins downstream of KDR

To identify KDR-dependent tyrosine phosphorylated proteins and specific phosphorylated tyrosine residues, MT-2 cells were treated with SU 1498 and leupeptin (to prevent Tax degradation) and lysates subjected to phosphoproteomics using PTMScan® (Cell Signaling Technology). Peptides phosphorylated on tyrosine residues were enriched with PTMScan® Phosphotyrosine pY-1000 Motif Antibody and subjected to LC-MS/MS analysis. A total of 491 tyrosine phosphorylated peptides were identified with significant upregulation or downregulation in response to KDR inhibition (Supplementary Fig. 7A and Supplementary Data 3). There was a significant downregulation of the tyrosine phosphorylation of components of the JAK-STAT pathway including JAK1, JAK2, JAK3, and STAT1 (Fig. 7C). The phosphorylation of MAP Kinases ERK1, ERK2, JNK1, JNK2, and p38 was also significantly decreased (Fig. 7C). To validate the phosphoproteomics results, specifically the JAK/STAT pathway, we performed immunoblotting to assess the phosphorylation of different effector molecules of JAK-STAT signaling (Fig. 7D, E). Indeed, there was diminished tyrosine phosphorylation of JAK1, JAK2, and JAK3 in MT-2 cells treated with SU 1498 and leupeptin (Fig. 7D). KDR inhibition, along with leupeptin treatment, also decreased the phosphorylated forms of STAT1 and STAT3 (Fig. 7E), suggesting compromised STAT activation in HTLV-1-transformed cells. In addition to JAK-STAT signaling, phosphoproteomics also identified altered tyrosine phosphorylation of multiple receptors/cell surface proteins, ubiquitin-conjugating systems, proteases, phosphatases, apoptotic regulatory factors, adhesion/extracellular matrix proteins as well as viral proteins (Rex, Gag-Pro and Env). (Supplementary Fig. 7B–E).

## KDR induces the phosphorylation of Tax

Given that Tax and KDR interact in cells, we considered the possibility that KDR may phosphorylate Tax on tyrosine residues; it should be noted that to our knowledge Tax tyrosine phosphorylation has not previously been described. To begin to test this hypothesis, we transiently transfected 293T cells with Tax and KDR plasmids, and immunoblotting was performed to examine the migration of Tax on SDS-

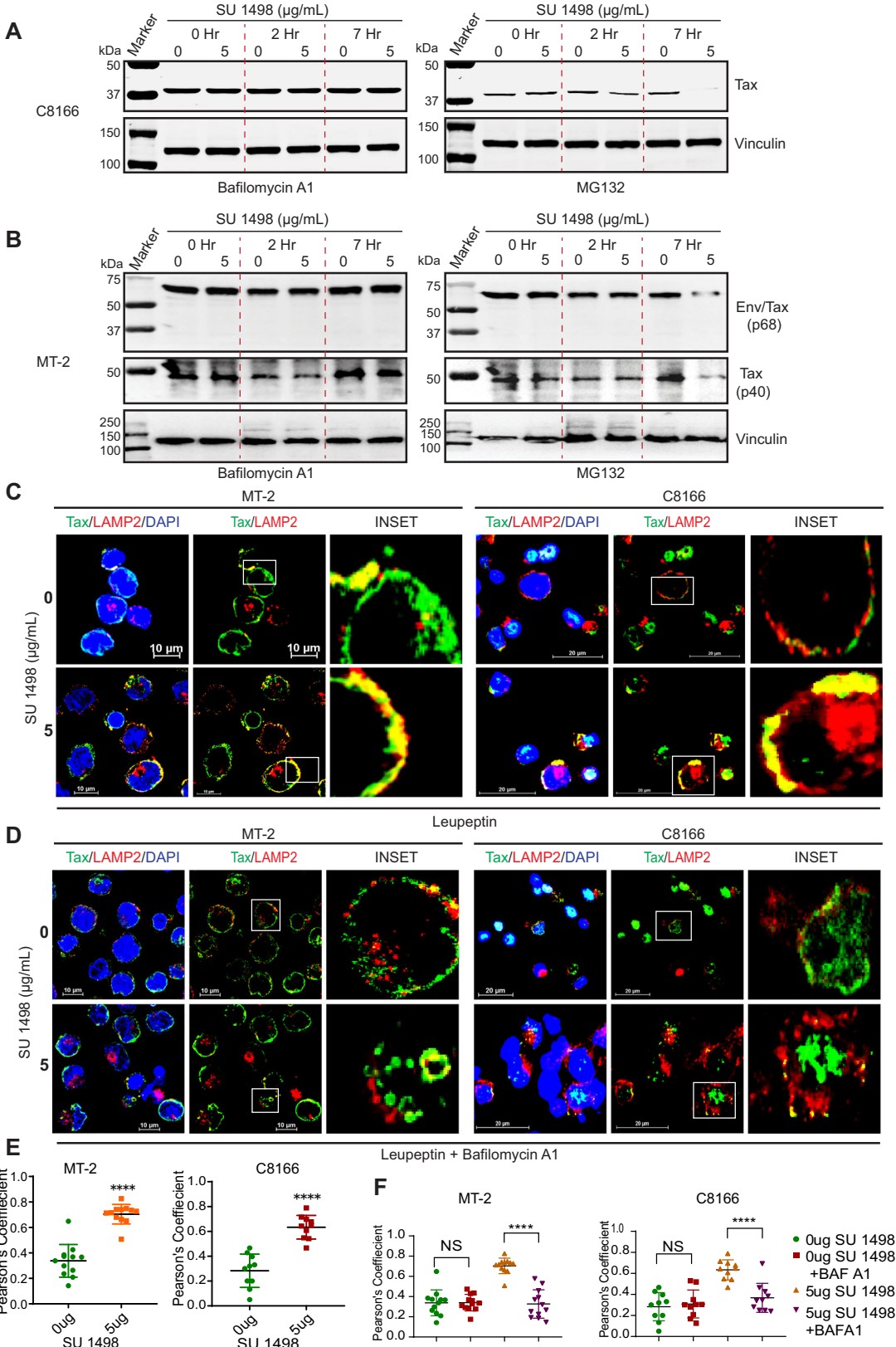

PAGE gels in the absence or presence of KDR. Interestingly, we observed a shift in the migration of Tax in the presence of KDR (Fig. 8A). To further substantiate these results, we used Phos-Tag gels which are based on phosphate-affinity electrophoresis whereby acrylamide is co-polymerized with Phos-Tag acrylamide, which acts as a phosphate-trapping molecule to separate phosphorylated and non-

phosphorylated proteins using conventional SDS-PAGE procedures (Fig. 8B). The advantage of this approach is that phosphorylated proteins undergo large mobility shifts on Phos-Tag gels. C8166 and MT-2 cells were treated with SU 1498 together with Bafilomycin A1 to prevent Tax degradation. A faster migrating, likely dephosphorylated, Tax band was detected by Phos-Tag gels, but not SDS-PAGE, in C8166 cells

**Fig. 3 | KDR inhibition induces autophagic/lysosomal degradation of Tax.** Immunoblotting was performed with the indicated antibodies using lysates from (**A**) C8166 and (**B**) MT-2 cells treated with 0 μg or 5 μg of SU 1498 at the indicated times. The experiments were independently repeated three times with similar results. Representative data from one experiment is shown. **C** Immunofluorescence confocal microscopy was performed using MT-2 and C8166 cells treated with SU 1498 and leupeptin (20 μM) for 24 h and labeled with Tax-Alexa Fluor 488 and LAMP2-Alexa Fluor 647 (pseudo red) antibodies and DAPI for nuclear staining. **D** Immunofluorescence confocal microscopy was performed using MT-2 and C8166 cells treated with SU 1498 and leupeptin (20 μM) and Bafilomycin A1 (20 nM) for 24 h and labeled with Tax-Alexa Fluor 488 and LAMP2-Alexa Fluor 647 antibodies and DAPI for nuclear staining. The experiments in (**C**, **D**) are representative of two independent experiments with similar results. **E**, **F** Graphical representation of Pearson's coefficient analysis of the number of cells showing localization of Tax in lysosomes (LAMP2), where each dot represents a single cell. MT-2 and C8166 cells were treated with SU 1498 and **E** leupeptin or **F** leupeptin (20 μM) and Bafilomycin A1 (20 nM) for 24 h. The results in (**E**) are expressed as the mean ± SD. Unpaired Student's $t$ test with Welch's correction. ****$P < 0.0001$. Two-tailed Student's $t$ test. P values from left to right: $P < 0.0001$, $P < 0.0001$ with $n = 12$ cells (MT-2) and $n = 10$ cells (C8166). The results in (**F**) are expressed as the mean ± SD. One-way ANOVA with Šídák's multiple comparisons test. ****$P < 0.0001$ and ns = not significant. P values from left to right: $P = 0.9991$, $P < 0.0001$, $P = 0.8735$, $P < 0.0001$. $n = 12$ cells (MT-2) and $n = 10$ cells (C8166). Source data are provided as a Source data file.

treated with SU 1498 and Bafilomycin A1 (Fig. 8C). Similar results were also obtained using MT-2 cells treated with SU 1498 and Bafilomycin A1 (Fig. 8D). To examine Tax tyrosine phosphorylation more directly, we performed co-IP assays using lysates from C8166 and MT-2 cells treated with SU 1498 and Bafilomycin A1. Tax was detected in anti-pTyr immunoprecipitates from C8166 and MT-2 cells but was diminished upon SU 1498 treatment (Fig. 8E, F). To determine if KDR directly phosphorylates Tax, we conducted an in vitro kinase assay with recombinant KDR and immunoprecipitated Tax from 293T cells transfected with Tax. After the in vitro kinase assay, the reactions were resolved on a Phos-Tag gel, followed by immunoblotting with anti-Tax. A shift in the migration of Tax was observed in the presence of KDR, confirming the phosphorylation of Tax by KDR (Fig. 8G). Together, these results indicate that KDR directly phosphorylates Tax.

### KDR inhibition induces Tax degradation and cell death and decreases p19 Gag expression in HAM/TSP PBMCs

Our collective results raise the intriguing possibility that KDR inhibitors could potentially be exploited to target Tax as a therapeutic strategy in HTLV-1-associated diseases. HAM/TSP patients have high proviral loads and express abundant levels of Tax, which drives neuroinflammation in these patients[39,40]. Therefore, KDR inhibitors could potentially be used to target Tax in HAM/TSP patients. To test this notion, we obtained PBMCs from patients with a clinical diagnosis of HAM/TSP harboring different HTLV-1 proviral loads and treated them with SU 1498 for 24 h. There was a significant induction of apoptosis of HAM/TSP PBMCs upon KDR inhibition in a concentration-dependent manner (Fig. 9A, C, D). The cell death was more pronounced in PBMCs from patients with low proviral loads, indicating an increased sensitivity to KDR inhibition. Importantly, KDR inhibition by SU 1498 exhibited little cytotoxicity in PBMCs derived from healthy donors (Fig. 9B–D), suggesting the specificity of KDR inhibition for HTLV-1-infected T cells from HAM/TSP patients. CD4+ and CD4 + CD25 + T cells serve as the major reservoir of the HTLV-1 proviral load. Flow cytometry-based immunostaining revealed a significant decline in the percentage of CD4+ and CD4 + CD25 + T cells upon KDR inhibition in HAM/TSP PBMCs, whereas no significant change was observed in CD8+ cells (Fig. 9E). Finally, immunoblotting was performed to evaluate the effect of KDR inhibition on the expression of Tax and p19 Gag in HAM/TSP PBMCs. Consistent with our earlier results with HTLV-1-transformed cell lines, KDR inhibition suppressed Tax and p19 Gag expression (Fig. 9F). These data suggest that KDR inhibition targets Tax and selectively depletes CD4 + T cells in HAM/TSP PBMCs.

## Discussion

Our study has identified KDR as an essential survival factor for HTLV-1-infected T cells expressing Tax. We have demonstrated that inhibition of KDR with either small-molecule inhibitors or shRNAs promotes the autophagic/lysosomal degradation of Tax and suppresses HTLV-1 replication and transmission. Tax interacts with KDR at the Golgi apparatus and is protected from degradation by KDR-mediated

tyrosine phosphorylation. Our results support a model (Fig. 10) whereby KDR inhibition induces the apoptotic cell death of HTLV-1-infected T cells through the degradation of Tax that disrupts the activation of oncogenic signaling pathways including NF-κB and JAK/STAT.

Tax is a potent viral trans-activator presumed to play a role in the early steps of ATLL leukemogenesis. ATLL cells typically exhibit no or low levels of Tax, which raises questions regarding its significance in maintaining the leukemic phenotype[41–43]. However, a recent study has highlighted the importance of Tax in HTLV-1 persistence and the survival of ATLL cells, which depends on a transient burst of Tax expression in a small fraction of leukemic cells[44]. Furthermore, low levels of Tax expression in primary ATLL cells are sufficient to maintain NF-κB activation, and inhibition of Tax expression triggers cell death, despite somatic mutations and expression of HBZ[8]. Although HTLV-1 is transcriptionally silent in freshly isolated cells from infected individuals, viral gene expression is induced by in vitro culture conditions and likely upregulated in vivo by stress stimuli such as hypoxia[45]. Therefore, Tax expression is transient in vivo, and potentially activated upon trafficking of infected cells to hypoxic environments such as lymph nodes and bone marrow. Nevertheless, the regulation of Tax expression and its stability by host proteins remains poorly understood. Our findings indicate a critical role for KDR in the sustained expression of the Tax protein in HTLV-1-transformed cells. Tax maintains its constitutive expression in HTLV-1-transformed cells by manipulating the autophagic machinery leading to the accumulation of autophagosomes but inhibition of the terminal step of autophagosome-lysosome fusion[17,18,46]. In this study, we found that Tax hijacks KDR kinase activity to prevent the autophagic/lysosomal degradation of Tax in HTLV-1-transformed cells. KDR inhibition enhances autophagic flux, specifically in Tax+ HTLV-1 transformed cells, leading to the degradation of Tax and cell death, indicating a disruption of Tax-regulated autophagic machinery. Furthermore, our results show that inhibiting KDR also blocks the replication and transmission of HTLV-1 in chronically infected HTLV-1-producing cell lines. Therefore, KDR inhibitors antagonize the cell survival of HTLV-1-infected T cells, viral gene expression, and HTLV-1 transmission, all key functions of Tax.

The pleiotropic function of Tax in activating aberrant cell signaling pathways depends on its PTMs and interactions with multiple host proteins[35]. In addition, Tax dynamically shuttles between multiple cellular compartments to perform specific functions[47,48]. For the persistent activation of NF-κB signaling, Tax interacts with the IKK complex at the cis-Golgi and manipulates transcriptional regulation within nuclear bodies[36,49,50]. The Golgi apparatus serves as a critical transport hub that supplies the necessary components for autophagosome formation to initiate autophagy[51], and also acts as a reservoir for inactivated forms of KDR[22]. We found an interaction between KDR and Tax at the Golgi apparatus, suggesting this interaction may regulate IKK activation and autophagosome biogenesis. Moreover, our data demonstrate that the relocalization of Tax to lysosomes upon KDR inhibition results in impaired NF-κB activation through the dissociation of the Tax-NEMO complex, indicating a critical role of KDR in Tax-

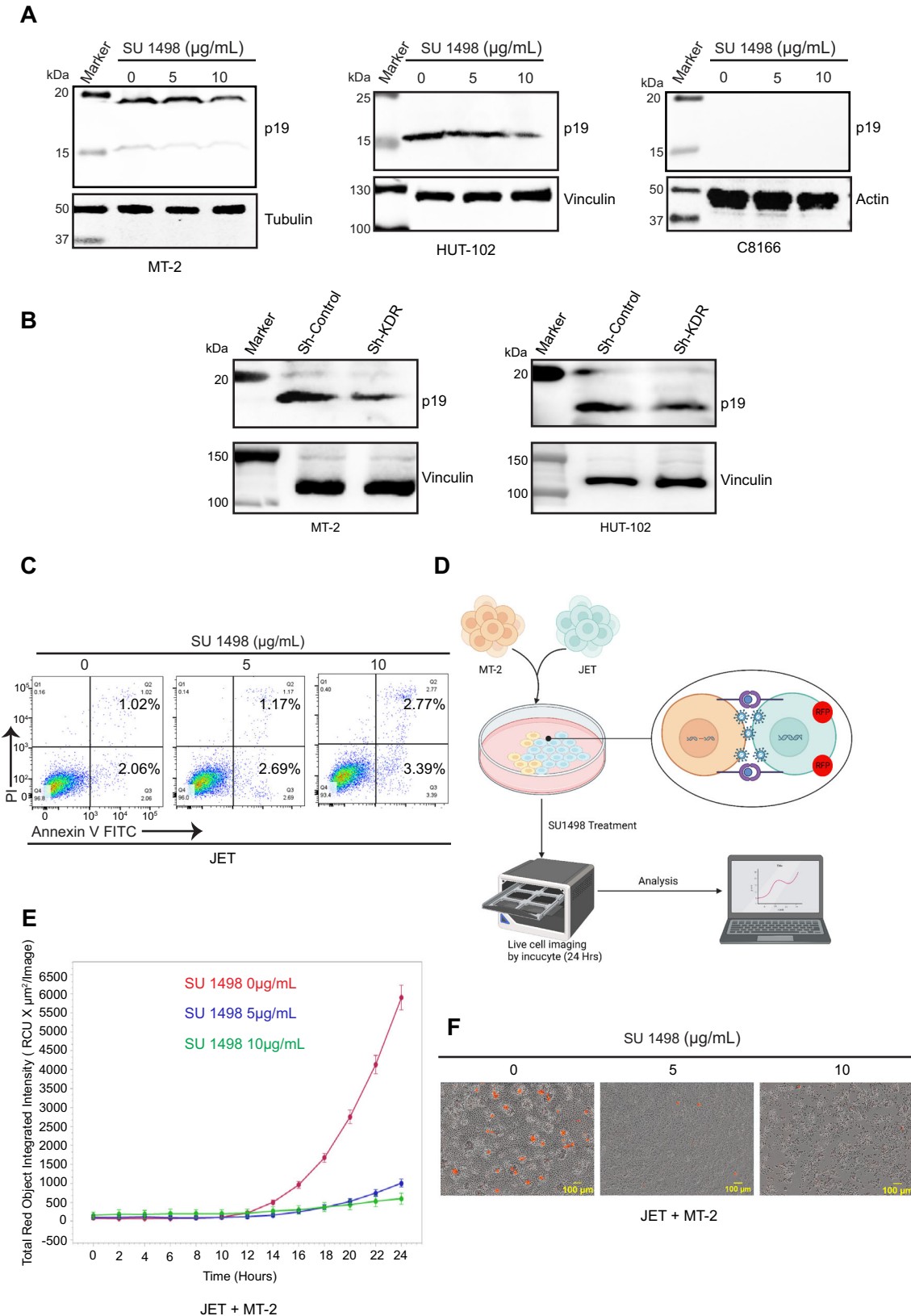

mediated NF-κB activation. Tax dysregulates cytokine production for sustained cell proliferation and immune evasion[10,11,52,53]. Our phosphoproteomics experiment demonstrated the downregulation of JAK-STAT signaling due to the relocalization of Tax to lysosomes upon KDR inhibition, resulting in the disruption of an established autocrine loop for the secretion of multiple cytokines.

As KDR is essential to maintain the stability of the Tax protein, Tax upregulates the mRNA and protein levels of KDR to ensure its stabilization. Previous studies have described a role for Tax in inducing VEGF secretion that promotes ATLL pathogenesis by linking angiogenesis with enhanced proviral load[54–56]. However, subsequent research has suggested Tax-independent secretion of VEGF[57,58], and no significant

**Fig. 4 | KDR inhibition inhibits HTLV-1 replication by suppressing p19 Gag expression and preventing HTLV-1 transmission. A** Immunoblotting was performed with p19 and vinculin antibodies using lysates from MT-2, HUT-102, and C8166 cells treated with SU 1498 for 24 h. The experiment is representative of two independent experiments with similar results. **B** Immunoblotting was performed with p19 and vinculin antibodies using lysates from MT-2 and HUT-102 cells expressing control scrambled shRNA or KDR shRNA. The experiment is representative of two independent experiments with similar results. **C** Annexin assay of JET cells treated with SU 1498 for 24 h and the percentage of pre-apoptotic (Annexin V⁺ and PI⁻) and post-apoptotic (Annexin⁺ and PI⁺) cell death was highlighted in representative pseudocolor plots demonstrating the gating strategy. **D** Schematic illustration of the co-culture of MT-2 and JET (1:3) cells and subsequent

treatment with SU 1498 for analysis by Incucyte S3 live-cell imaging of RFP expression in infected JET cells. The schematic was created with BioRender.com and released under a Creative Commons Attribution-NonCommercial-NoDerivs 4.0 International license. **E** Incucyte S3 live-cell imaging of RFP expression at 2-h intervals in co-cultured MT-2 and JET cells treated with SU 1498 for 24 h. At each time point, images were taken from 24 wells (8 wells/sample and 16 images/well) in the phase bright-field and red channel at ×10 magnification with an S3/SX1 G/R optical module. The error bars are presented as a standard error with *n* = 8 technical replicates. The experiment is representative of two independent experiments with similar results. **F** Representative images of Incucyte S3 live-cell analysis of co-cultured MT-2 and JET cells treated with SU 1498 for 24 h with ×10 magnification. Source data are provided as a Source data file.

differences were found in VEGF levels between HAM/TSP, ATLL, and healthy or asymptomatic carriers[59,60]. Furthermore, a recent study revealed a lack of efficacy of intraocular anti-VEGF antibody treatment on activated NF-κB, inflammatory cytokines/chemokines, and HTLV-1 proviral load[61]. The VEGF-A splice variant VEGF$_{165}$ is a ligand of the HTLV-1 receptor Neuropilin-1 (NRP-1), and HTLV-1 mimics VEGF$_{165}$ to recruit HSPGs and NRP-1[62]. Together, these studies indicate complex roles for VEGF in HTLV-1 infection and HTLV-1-associated pathogenesis, and further studies are needed to elucidate its functional roles. Nevertheless, our data suggest that VEGF is unlikely to exert an effect on KDR-mediated Tax stability since Tax sequesters KDR in a perinuclear region where it is unable to interact with any VEGF ligands.

Previous studies have identified serine/threonine phosphorylation sites in Tax[63,64], and have further established the significance of serine phosphorylation of Tax in regulating its nuclear translocation and activation of CREB and NF-κB pathways[65]. However, tyrosine phosphorylation of Tax has not yet been reported. Since KDR is a tyrosine kinase interacting with Tax to regulate its stability, we examined potential phosphorylation of Tax by KDR. Indeed, our in vitro kinase assay indicates that Tax is directly phosphorylated by KDR, presumably on tyrosine residues. How Tax phosphorylation by KDR protects Tax from degradation by autophagosomes/lysosomes remains an open question. We speculate that Tax phosphorylation by KDR may inhibit its recognition and interaction with selective autophagy receptors that target Tax to autophagosomes. Alternatively, Tax tyrosine phosphorylation by KDR may promote an interaction with host factors that regulate autophagosome-lysosome fusion. These potential mechanisms will be addressed in future studies.

Accumulating clinical trials have indicated poor efficacy of chemotherapy in treating ATLL patients due to the lack of specificity and targeted action towards viral proteins[66]. Antiviral approaches based on zidovudine and IFN-α showed high response rates in clinical trials; however, the survival rate of patients decreased when administered with chemotherapy[67,68]. Treatment of ATLL cells with arsenic trioxide and IFN-α triggered the proteasomal degradation of Tax and cell death, suggesting that a therapeutic approach targeting Tax may indeed improve outcomes in ATLL patients[69]. Although recent treatment modalities for ATLL such as the immunomodulator lenalidomide, anti-CCR4 antibody mogamulizumab, histone deacetylase inhibitor tucidinostat, and histone methyltransferase EZH1/2 dual inhibitor valemetostat are available in Japan for the treatment of ATLL, these treatment options lack antiviral approaches to suppress HTLV-1 replication and transmission[70–73]. The Tax peptide-pulsed dendritic cell vaccine demonstrated therapeutic effectiveness in treating Tax-positive ATLL patients[74]. Therefore, therapeutic approaches that combine antiviral and chemotherapeutic approaches may be more effective in treating patients with ATLL. Although HAM/TSP patients harbor a high HTLV-1 proviral load with elevated Tax expression, current therapies for HAM/TSP patients mainly focus on treating clinical symptoms with anti-inflammatory corticosteroids, and inducing viral gene expression to increase the susceptibility of infected cells to immune responses by treatment with valproate, a histone deacetylase

inhibitor[75,76]. Nevertheless, current treatment options for HAM/TSP patients are largely ineffective. Our results suggest that targeting Tax with KDR inhibitors may potentially serve as a feasible therapeutic strategy for HAM/TSP patients. KDR inhibition led to significant cell death of CD4+ and CD4 + CD25 + T cells from HAM/TSP patients as well as Tax degradation and decreased p19 Gag expression, indicating impaired HTLV-1 replication. Altogether, our results provide a strong rationale for the clinical investigation of KDR inhibitors for patients with HAM/TSP and Tax+ ATLL. Notably, several small-molecule inhibitors and monoclonal antibodies targeting KDR are already approved by the FDA for cancer therapy. Repurposing of these drugs for HAM/TSP and/or ATLL patients should be considered for clinical trials.

In conclusion, we identified KDR by a kinome-wide shRNA screen and validated its critical role in the survival of HTLV-1-infected T cells. We found that KDR binds to and phosphorylates Tax to prevent its degradation. Overall, our study has revealed that Tax stability is controlled by KDR kinase activity which could be exploited as a strategy to target Tax in ATLL and/or HAM/TSP.

## Methods

### Ethics statement
Blood samples from HAM/TSP patients were collected under protocol# 98N0047 approved by the National Institutes of Health IRB #10 (the NIH Intramural IRB), IRB Registration: IRB00011862 and the National Institute of Neurologic Disorders and Stroke (NINDS) Scientific Review Committee. Prior to study inclusion, written informed consent was obtained from subjects in accordance with the Declaration of Helsinki.

### Cell culture, plasmid, and antibodies
HTLV-1 transformed T-cell lines C8166, HUT-102, and SLB-1 were obtained from Dr. Shao-Cong Sun[77]. TL-OM1 and MT-1 cell lines were provided by Dr. Michiyuki Maeda[78]. ATL-2S cells were provided by Dr. Masao Matsuoka[30]. HTLV-1 reporter JET cells were a gift from Dr. Jun-ichi Fujisawa[29]. Jurkat Tax Tet-On cells were provided by Dr. Warner Greene[79]. Human embryonic kidney cells (HEK 293T; CRL-3216), Jurkat (TIB-152) and HL-60 (CCL-240) were purchased from ATCC. MT-2 cells were obtained from the AIDS Research and Reference Program, NIAID, National Institutes of Health (currently BEI Resources; catalog#ARP-237). Jurkat clone E6-1, JET, Jurkat Tax Tet-On, C8166, MT-2, HUT-102, SLB-1, MT-1, ATL-2S and TL-OM1 cells were cultured in RPMI medium. HEK 293T cells were cultured in Dulbecco's modified Eagle's medium (DMEM). The medium was supplemented with fetal bovine serum (10%) and penicillin−streptomycin (1%). Jurkat Tax Tet-On cells were cultured in RPMI medium with Tet System Approved FBS (Takara). PBMCs obtained from three patients with a clinical diagnosis of HAM/TSP and harboring different HTLV-1 viral loads (Supplementary Table 1), were cultured in RPMI medium supplemented with fetal bovine serum (10%) and penicillin−streptomycin (1%) for five days (to induce Tax expression) prior to drug treatment. PBMCs from healthy donors were cultured in RMPI medium for two days and examined for cell viability following treatment with KDR inhibitors. The pCMV4-Tax

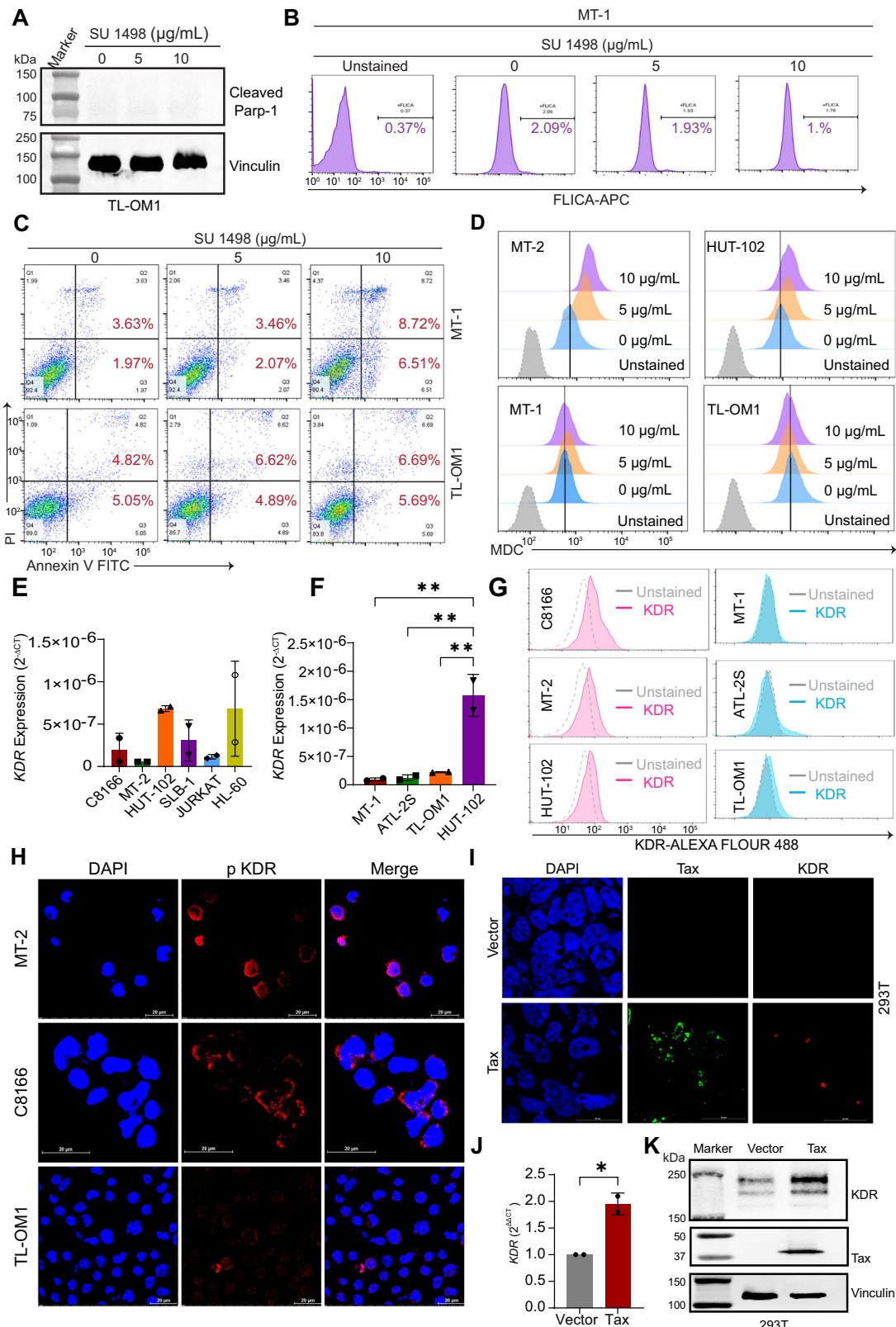

plasmid was provided by Dr. Shao-Cong Sun[80]. hKDR (GFP-KDR) was a gift from Christie Thomas (Addgene plasmid #83436; http://n2t.net/addgene:83436; RRID: Addgene_83436). Anti-Tax (1A3; sc-57872), NEMO (sc-8032), KDR (sc-6251), vinculin (sc-73614), and β-actin (sc-47778) antibodies were purchased from Santa Cruz Biotechnology. Cleaved PARP-1 (#5625), pIκBα (#9246), pIKKα/β (#2697), IKKβ (#8943), IκBα (#4812), KDR (#2479), JAK1 (#3344), JAK2 (#3230), JAK3

(#8827), pJAK1 (#74129), pJAK2 (#8082), pJAK3 (#5031), STAT1 (#14994), STAT3 (#4904), pSTAT1 (#7649), pSTAT3 Tyr705 (#9145), pSTAT3 Ser727 (#9134) and pTyrosine (#9411) were purchased from Cell Signaling Technology. The p19 Gag antibody (#801003) was purchased from ZeptoMetrix. Rabbit IgG HRP Linked Whole Ab (NA934-1ML) and Mouse IgG HRP Linked Whole Ab (NA931-1ML) were purchased from Cytiva. GM-130 polyclonal antibody, Alexa Fluor 647 (PA1-

**Fig. 5 | Tax induces KDR expression. A** Immunoblotting was performed with cleaved PARP-1 and Vinculin antibodies using lysates from TL-OM1 cells treated with SU 1498 for 24 h. The experiment is representative of three independent experiments with similar results. **B** FLICA caspase assay was performed using MT-1 cells treated with SU 1498 for 24 h. The histograms indicate the percentage of active caspase 3/7 in treated cells. **C** Annexin assay of MT-1 and TL-OM1 cells treated with SU 1498 for 24 h and the percentage of pre-apoptotic (Annexin V+ and PI-) and post-apoptotic (Annexin V+ and PI+) cell death was shown in representative pseudocolor plots demonstrating the gating strategy. **D** Tax+ (MT-2 and HUT-102) and Tax- (MT-1 and TL-OM1) cells were treated with SU 1498 for 24 h and stained with MDC for acquisition by flow cytometry. Histogram overlays show changes in MDC staining levels in treated cells, indicating autophagy induction. **E** qRT-PCR of KDR mRNA in C8166, MT-2, HUT-102, SLB-1, Jurkat, and HL-60 cells. The results are expressed as the mean ± SD from two independent experiments. **F** qRT-PCR of KDR mRNA in MT-1, ATL-2S, TL-OM1, and HUT-102 cells. The results are expressed as the mean ± SD

from two independent experiments. **P < 0.01. One-way ANOVA with Dunnett's multiple comparisons test with P values from left to right: $P = 0.0032$, $P = 0.0035$, $P = 0.0045$. **G** Flow cytometry-based analysis of KDR protein expression in Tax+ and Tax- ATLL cell lines. **H** Immunofluorescence confocal microscopy was performed using MT-2, C8166, and TL-OM1 cells labeled with pKDR-Alexa Fluor 488 and DAPI for nuclear staining. **I** Immunofluorescence confocal microscopy was performed using 293T cells transiently transfected with Tax and stained with Tax-Alexa Fluor 488, KDR-Alexa Fluor 594, and DAPI for nuclear staining. The experiments in (**H**, **I**) are representative of two independent experiments with similar results. **J** qRT-PCR of KDR mRNAs in 293T cells transiently transfected with Tax. The results are expressed as the mean ± SD from two independent experiments. *P < 0.05. Two-tailed Student's *t* test. $P = 0.0228$. **K** Immunoblotting was performed using the indicated antibodies in 293T cells transiently transfected with Tax. The experiment is representative of two independent experiments with similar results. Source data are provided as a Source data file.

077-A647), LAMP2 monoclonal antibody, Alexa Fluor 647 (A15464), Alexa Fluor 594-conjugated donkey anti-mouse IgG (A-11005) and Alexa Fluor 488-conjugated donkey anti-rabbit IgG (A-11008) were purchased from Thermo Fisher Scientific. Human TruStain FcX (422301), Alexa Fluor 594 anti-human CD4 (300544), Alexa Fluor® 594 Mouse IgG1, κ Isotype Ctrl (400174), APC/Fire 750 anti-human CD3 (317352), APC/Fire 750 mouse IgG2a, κ isotype control (400284), Brilliant Violet 421 anti-human CD25 (302630), Brilliant Violet 421™ Mouse IgG1, κ Isotype (400158), Brilliant Violet 711™ Mouse IgG1, κ Isotype Ctrl (400168) and Brilliant Violet 711 anti-human CD8a (301044) were purchased from BioLegend. DAPI (4′, 6-diamidino-2-phenylindole) was purchased from EMD Biosciences. SU 1498, SKLB 1002, and MDC were purchased from Millipore-Sigma. MitoTracker™ Deep Red and SuperSignal West Pico Chemiluminescent reagents were purchased from Thermo Fisher Scientific. Doxycycline was purchased from Takara. Puromycin was from Thermo Fisher Scientific. Cabozantinib, Mem-PER Plus membrane protein extraction kit, Annexin V Alexa Fluor 488 and Annexin V binding buffer were purchased from Thermo Fisher Scientific.

### Cell viability and proliferation assays
Cell viability was determined using the CellTiter-Glo luminescent cell viability assay (Promega), which quantifies ATP as a measure of metabolically active cells. A total of 50 µl of suspended cells and 50 µl of CellTiter-Glo solution were mixed and incubated at room temperature for 10 min, and the luminescence was quantified with a Glo-Max96 microplate luminometer (Promega).

### Immunoblotting and co-immunoprecipitation assays
Whole-cell lysates were generated by lysing cells in RIPA buffer (50 mM Tris-Cl [pH 7.4], 150 mM NaCl, 1% NP-40, 0.25% sodium deoxycholate, Pierce Protease and Phosphatase Inhibitor) on ice, followed by centrifugation. Cell lysates were resolved by SDS-PAGE, transferred to nitrocellulose membranes using the Trans-Blot Turbo Transfer System, and subjected to immunoblotting with the indicated primary antibodies and HRP-conjugated secondary antibodies (Cytiva Life Sciences). Immunoreactive bands were visualized using SuperSignal West Pico Chemiluminescent reagent and analyzed with a Bio-Rad ChemiDoc or Azure 600 Western Blot Imaging System. Western blot images were processed using Image Lab software 6.1 (Bio-Rad Laboratories). The Dynabeads Protein G Immunoprecipitation Kit (Thermo Fisher Scientific; #10007D) was used for co-IP assays according to the manufacturer's recommendations.

### Annexin V apoptosis detection and cell cycle assay
Cells were harvested, washed with PBS, stained with Annexin V-FITC according to the manufacturer's recommended concentration and PI (2 µg) in 1x binding buffer for 30 min, and acquired using a FACS Symphony A3 flow cytometer (Becton-Dickinson). Analysis was

performed using FlowJo v10 software (Tree Star, USA). For cell cycle staining, cells were washed with PBS, fixed with ice-cold 70% ethanol on ice for 30 min, and washed with PBS followed by staining with PI (10 µg/mL) in 0.05% of Triton X-100 containing RNase (100 µg/mL) for 30 min at room temperature in the dark. Data was acquired using a FACS Calibur flow cytometer (Becton-Dickinson) and analyzed by FlowJo v10 software (Tree Star, USA)[81]. Only signals from single cells were considered (10,000 cells/sample) for analysis. The gating strategy for these experiments is provided in Supplementary Figs. 8–10.

### FLICA caspase assay
The caspase 3/7 FLICA kit was purchased from Bio-Rad. Cells were stained according to the manufacturer's protocol and acquired on a BD FACS Symphony A3 flow cytometer. Data analysis was performed using FlowJo v10 software. The gating strategy for these experiments is provided in Supplementary Fig. 11.

### Autophagolysosome detection assay
Cells were stained with 50 µM MDC (monodansylcadaverine) for 30 min at 37 °C, and the intensity of violet fluorescence was acquired on a BD FACS Symphony A3 flow cytometer and analyzed using FlowJo v10 software. The gating strategy for these experiments is provided in Supplementary Fig. 12.

### Intracellular staining for flow cytometry
Cells were labeled with appropriate concentrations of primary antibodies, followed by staining with fluorescent-conjugated secondary antibodies as recommended by the eBioscience intracellular staining protocol and acquired on a BD FACS Symphony A3 flow cytometer. Data analysis was performed using FlowJo v10 software. The gating strategy for these experiments is provided in Supplementary Fig. 13.

### Confocal microscopy
Cells were washed with PBS and seeded in poly L-lysine pre-coated coverslips. Cells were fixed with 4% paraformaldehyde for 30 min and permeabilized with Triton X-100 for 5 min. Fixed cells were incubated with 5% BSA for 1 h followed by staining with primary antibodies overnight at 4 °C and fluorescently-conjugated secondary antibodies for 1 h at room temperature[82]. DAPI was used to stain nuclei. GM-130 and LAMP2 antibodies conjugated to Alexa Fluor 647 were incubated for 1 h at room temperature. Images were acquired with a C2+ confocal microscope system (Nikon) and processed using NIS-Elements software.

### Proximity ligation assay
PLA was performed using the Duolink® In Situ Red Starter Kit Mouse/Rabbit (Millipore-Sigma) as recommended by the manufacturer.

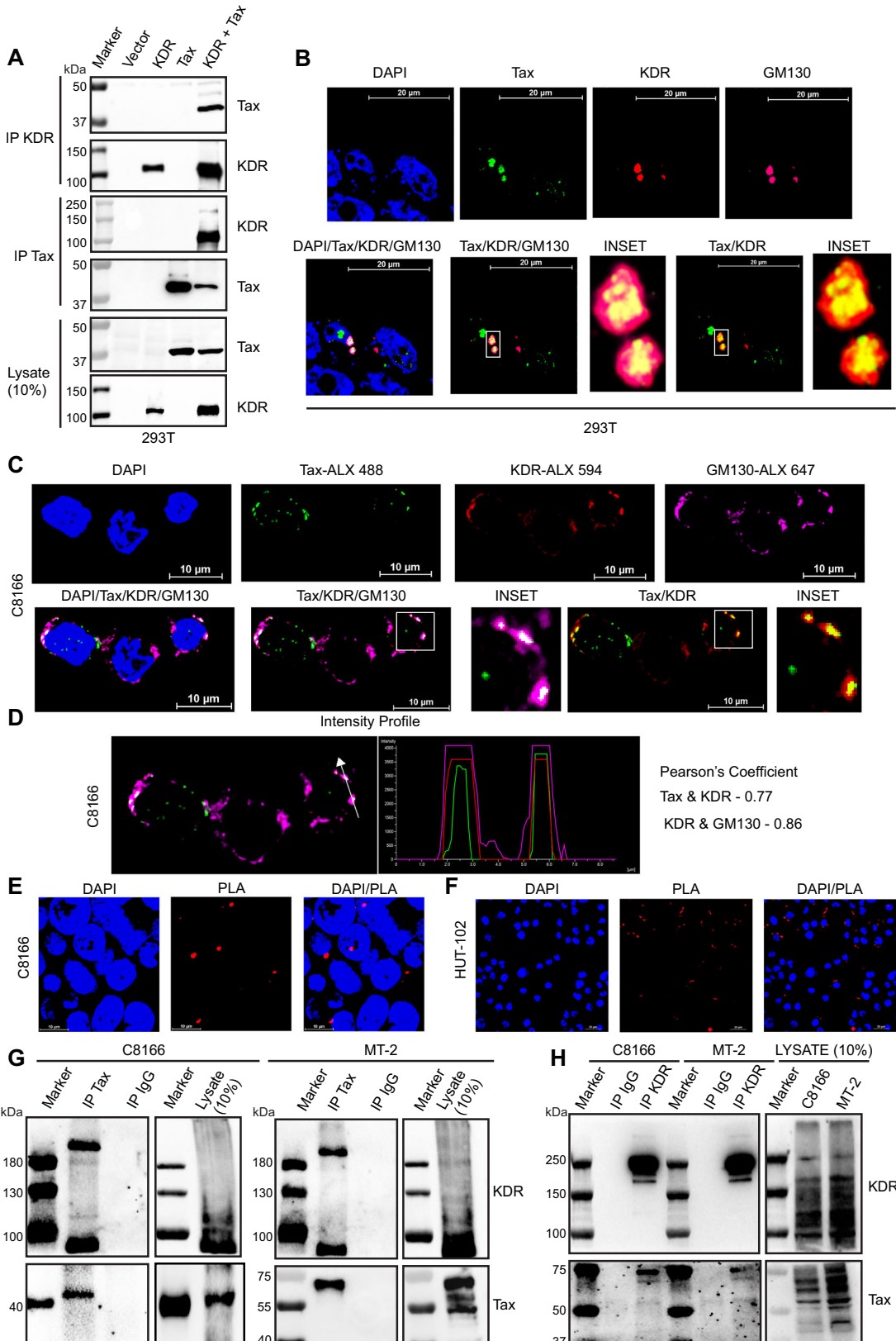

**Fig. 6 | Tax interacts with and colocalizes with KDR at the Golgi. A** Co-IP analysis of Tax and KDR-GFP from lysates of transfected 293T cells. Immunoblotting was performed with lysates using the indicated antibodies. **B** Immunofluorescence confocal microscopy was performed using 293T cells stained with the indicated antibodies. **C** Immunofluorescence confocal microscopy was performed using C8166 cells with the indicated antibodies. **D** The fluorescence intensity profile is plotted along the white arrow represented in the graph showing overlap, and Pearson's coefficient analysis was performed using NIS-Element software. **E, F** PLA was performed using C8166 (**E**) and HUT-102 (**F**) cells with Tax and KDR antibodies. The cells were labeled with DAPI for nuclear staining. **G, H** Co-IP assay was performed with either control IgG, anti-Tax (**G**) or anti-KDR (**H**) immunoprecipitates from lysates of C8166 and MT-2 cells as indicated. The experiments in (**A**–**H**) are representative of two independent experiments with similar results. Source data are provided as a Source data file.

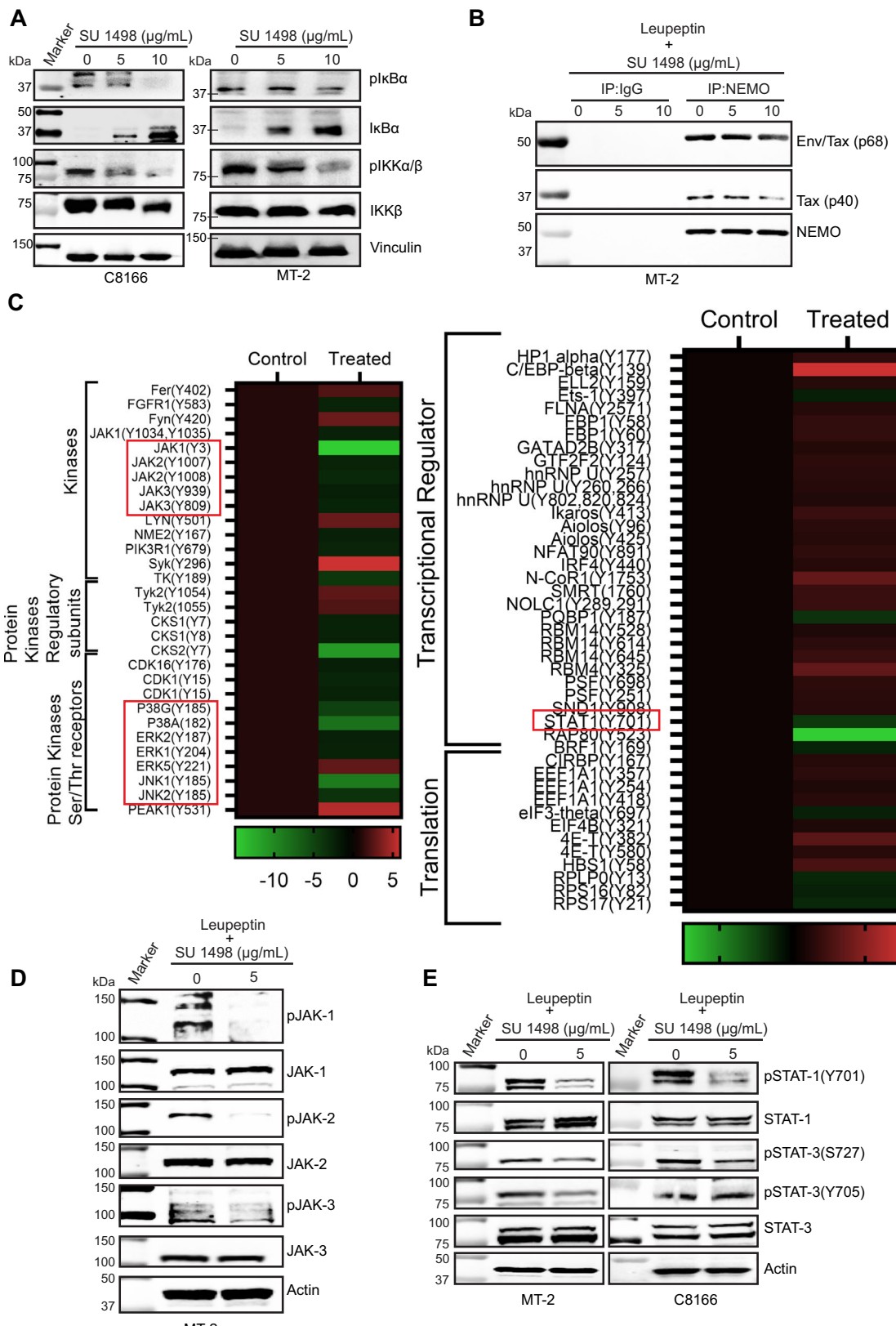

**Fig. 7 | KDR inhibition impairs NF-κB and JAK/STAT signaling pathways.**
**A** Immunoblotting was performed with the indicated antibodies using lysates from C8166 and MT-2 cells treated with SU 1498 for 24 h. The experiment is representative of two independent experiments with similar results. **B** Co-IP assay was performed with either control IgG or anti-NEMO using lysates from MT-2 cells treated with SU 1498 and leupeptin (20 µM) for 24 h. The experiment is representative of two independent experiments with similar results. **C** Heatmap representation of phosphoproteomics results in SU 1498-treated MT-2 cells showing alterations in tyrosine phosphorylation of proteins grouped according to their functions. The color gradient indicates the intensity of gene expression.
**D**, **E** Immunoblotting was performed with the indicated antibodies specific for JAK-STAT signaling components using lysates from MT-2 and C8166 cells treated with SU 1498 and leupeptin (20 µM) for 24 h. The experiments were independently repeated two times with similar results. Representative data from one experiment is shown. Source data are provided as a Source data file.

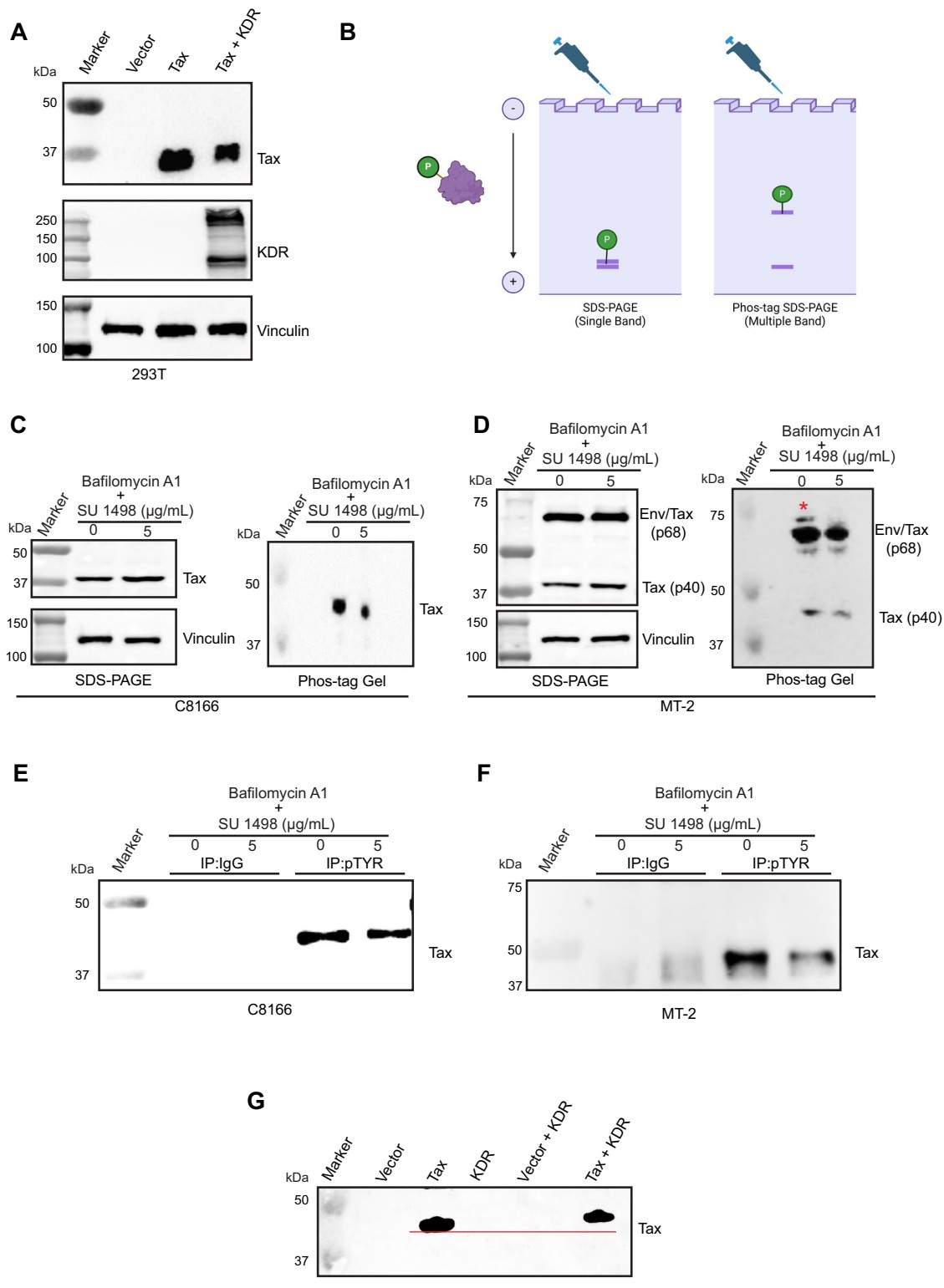

**Fig. 8 | KDR induces Tax phosphorylation. A** Immunoblotting was performed with the indicated antibodies using lysates from 293T cells transfected with Tax and/or KDR-GFP plasmids. The experiment is representative of two independent experiments with similar results. **B** Schematic depicting the principle of Phos-Tag gels for the analysis of phosphorylated proteins. The schematic was created with BioRender.com and released under a Creative Commons Attribution-NonCommercial-NoDerivs 4.0 International license. **C, D** Lysates from C8166 (**C**) and MT-2 cells (**D**) treated with SU 1498 and Bafilomycin A1 (20 nM) for 24 h were resolved by SDS-PAGE and Phos-Tag gels and subjected to immunoblotting. **E, F** Co-IP assay was performed with either control IgG or phosphoTyrosine (pTYR) immunoprecipitates from lysates of C8166 (**E**) or MT-2 cells (**F**) treated with SU 1498 and Bafilomycin A1 (20 nM) for 24 h. **G** In vitro kinase assay using immuno-precipitated Tax and recombinant KDR. The reaction was resolved on a Phos-Tag gel and subjected to immunoblotting. The experiments in (**C–G**) were independently repeated two times with similar results. Representative data from one experiment is shown. Source data are provided as a Source data file.

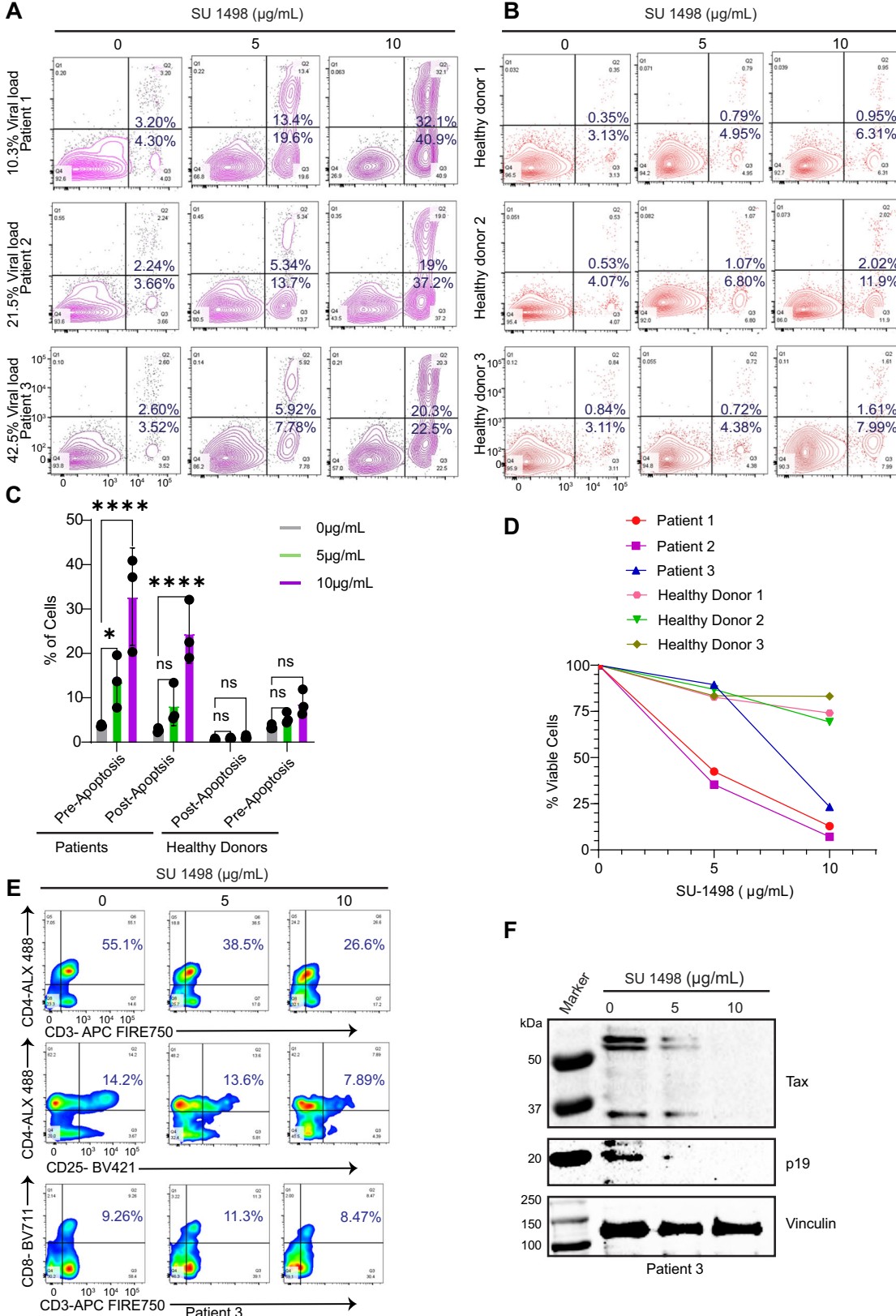

C8166 and HUT-102 cells were grown on poly L-lysine pre-coated coverslips, fixed, permeabilized, and incubated with primary antibodies anti-KDR (rabbit) and anti-Tax (mouse). The slides were incubated with Duolink® PLA probes, ligated, amplified, and washed according to the manufacturer's protocol. Images were acquired with a C2+ confocal microscope system (Nikon) and processed using NIS-Elements software.

**Mitochondrial membrane potential assay**

Cells were washed and stained with 140 nM MitoTracker Deep Red. The cells were then incubated for 45 min at 37 °C in the dark and washed 3× with PBS. The cells were fixed with 4% paraformaldehyde for 30 min and mounted with Prolong Gold Antifade Reagent with DAPI. Images were acquired with a C2+ confocal microscope system (Nikon) and processed using NIS-Elements software.

**Fig. 9 | KDR inhibition induces Tax degradation and cell death and decreases p19 Gag expression in HAM/TSP PBMCs. A**, **B** Annexin assay of HAM/TSP PBMCs (**A**) and healthy donor PBMCs (**B**) treated with SU 1498 for 24 h and the percentage of pre-apoptotic (Annexin V⁺ and PI⁻) and post-apoptotic (Annexin V⁺ and PI⁺) cell death is indicated in representative pseudocolor plots demonstrating the gating strategy. **C** Graphical representation of the percentage of pre-apoptotic and post-apoptotic cells in HAM/TSP PBMCs and healthy donor PBMCs treated with SU 1498 for 24 h. ****$P < 0.0001$; *$P < 0.05$. Two-way ANOVA with Dunnett's multiple comparisons test. $P$ values from left to right: $P = 0.0198$, $P < 0.0001$, $P = 0.2322$, $P < 0.0001$, $P = 0.9953$, $P = 0.9824$, $P = 0.8125$, $P = 0.2594$, ns not significant. Data

are represented as mean ± SD from three biological replicates (HAM/TSP patients and healthy donor PBMCs). **D** Cell viability assay was performed using HAM/TSP PBMCs, and healthy donor PBMCs treated with SU 1498 for 24 h. **E** Flow cytometry-based analysis of the percentage of CD3 + CD4 + CD25+ in HAM/TSP PBMCs treated with SU 1498 for 24 h. **F** Immunoblotting was performed with the indicated antibodies using lysates from HAM/TSP PBMCs (patient#3) treated with SU 1498 for 24 h. The experiment was repeated two times with technical replicates with similar results. Representative data from one experiment is shown. Source data are provided as a Source data file.

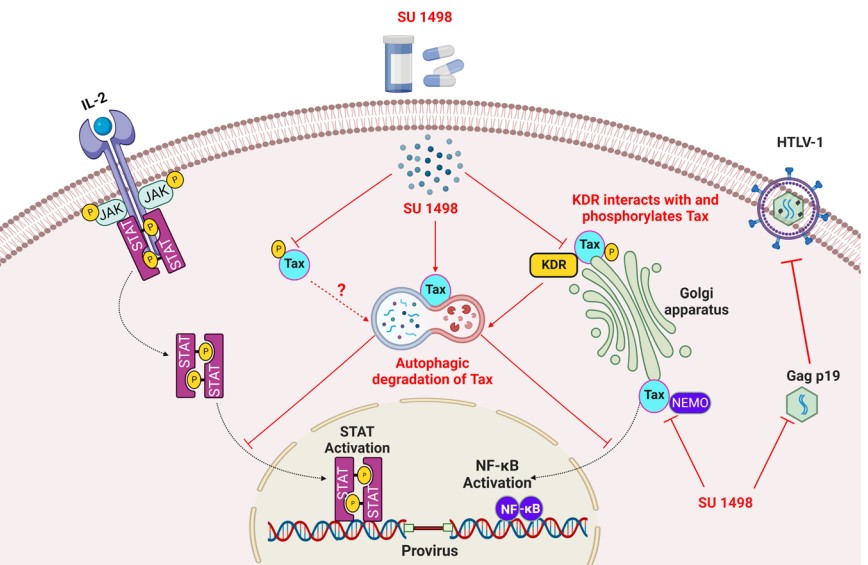

**Fig. 10 | Model depicting the role of KDR in protecting Tax from autophagic degradation.** Tax induces the expression of KDR and interacts with KDR at the Golgi. KDR phosphorylates Tax and protects Tax from autophagic degradation. KDR inhibitors such as SU 1498 trigger the degradation of Tax which leads to the downregulation of prosurvival NF-κB and JAK/STAT pathways. KDR inhibition also leads to decreased viral gene expression, disruption of the Tax-NEMO complex and the targeted cell death of Tax-expressing HTLV-1-infected T cells. The schematic was created with BioRender.com and released under a Creative Commons Attribution-NonCommercial-NoDerivs 4.0 International license.

## Plasmid and siRNA transfections

293T cells were transiently transfected with plasmids using GenJet In Vitro DNA Transfection Reagent (SignaGen Laboratories). Control ON-Target plus nontargeting siRNA #1 (D-001810-01-05) and ON-Target plus human KDR siRNA, SMARTpool (L-003148-00-0005) were purchased from Horizon Discovery. C8166 cells were transfected with siRNAs using Lipofectamine RNAiMAX transfection reagent (Thermo Fisher Scientific), according to the manufacturer's protocol.

## TaqMan real-time PCR assays

RNA was isolated using the RNeasy Mini Kit (Qiagen, Hilden, Germany). RNA was converted to cDNA using the First Strand cDNA synthesis kit for reverse transcription (avian myeloblastosis virus [AMV]; Millipore-Sigma). The TaqMan RT-PCR reaction was performed with a KDR probe (Hs00911708_m1) and TaqMan reaction mixture according to the manufacturer's protocol using a QuantStudio 3 Real-Time PCR system. Gene expression was normalized to the GAPDH (Hs03929097_g1) internal control. TaqMan probes were purchased from Thermo Fisher Scientific.

## Kinome-wide shRNA screen

The kinome-wide shRNA screen was performed with the MISSION® LentiExpress Kinome RNAi screening platform (Millipore-Sigma) consisting of pre-arrayed lentiviral particles in 41 × 96-well plates. There was a total of ~3200 lentiviruses targeting 501 human kinases. On average there were 5 shRNA constructs per gene target with 4 shRNAs within the coding sequence of the gene and 1 shRNA within the 3′ UTR

region. Negative controls (empty vector+nontargeting shRNA) were included in each 96-well plate. MT-2 cells were dispensed into the 96-well lentiviral shRNA plates, and then selected with puromycin for 3 days followed by assessment of cell viability using CellTiter-Glo® (Promega).

## Knockdown with lentiviral shRNAs

The SMARTvector KDR shRNA hCMV-TurboGFP plasmid (V3SH11240-225708235) and SMARTvector nontargeting hCMV-TurboGFP (VSC11707) were purchased from Horizon Discovery. Lenti-X™ 293T cells were transfected with Lenti-X packaging single shots (Takara) and lentiviral supernatants were collected 72 h post-transfection. The supernatants were centrifuged at 500×$g$ for 10 min to remove cell debris and then concentrated using a Lenti-X™ concentrator (Takara). The concentrated lentiviral stocks were quantified using the Lenti-X™ qRT-PCR titration kit (Takara) and transduced (MOI = 25) into C8166 and MT-2 cells.

## Phosphoproteomics

The PTMScan® Discovery Proteomics service was performed by Cell Signaling Technology. MT-2 cells were treated with vehicle or SU 1498 (5 µg/ml) + leupeptin (20 µM) for 24 h. There was a total of 1 biological replicate (two samples) with replicate injections of each run non-sequentially. Cells were lysed in Urea Lysis Buffer containing 20 mM HEPES (pH 8.0), 9.0 M Urea, 1 mM sodium orthovanadate (activated), 2.5 mM sodium pyrophosphate and 1 mM β-glycerol-phosphate. Cell lysates were sonicated 3 × 15 s at 15 W output power

using a probe sonicator with ~1 min on ice between bursts, centrifuged at 20,000×*g* for 15 min and the supernatant was transferred to a new tube. Lysates were digested with trypsin and loaded directly onto a 50 cm × 100 mm PicoFrit capillary column packed with C18 reversed-phase resin[83]. The column was developed with a 90 min linear gradient of acetonitrile in 0.125% formic acid delivered at 280 nL/min. The peptides were enriched by immunoprecipitating with PTMScan® Phosphotyrosine pY-1000 Motif Antibody (Cell Signaling Technology; #38572) immobilized to Protein A/G agarose. Peptides were eluted and subjected to LC-MS/MS using an Orbitrap-Fusion Lumos, ESI-HCD. The MS Parameter settings were MS Run Time 108 min, MS1 Scan Range (300.0–1500.00), 3 s cycle time MS/MS (Min Signal 500, Isolation Width 2.0, Normalized Coll. Energy 35.0, Activation-Q 0.250, Activation Time 20.0, Lock Mass 371.101237, Charge State Rejection Enabled, Charge State 1+ Rejected, Dynamic Exclusion Enabled, Repeat Count 1, Repeat Duration 35.0, Exclusion List Size 500, Exclusion Duration 40.0, Exclusion Mass Width Relative to Mass, Exclusion Mass Width 10 ppm). MS/MS spectra were evaluated using Comet[84] and the GFY-Core platform version 3.8 from Harvard University. Searches were performed against the most recent update of the Uniprot *Homo Sapiens* database with mass accuracy of +/−50 ppm for precursor ions and 0.02 Da for product ions. Results were filtered with mass accuracy of +/−5 ppm on precursor ions and the presence of the intended motif. A standard target/decoy strategy with a 1% peptide level FDR was used to filter the data using the linear discriminant module in GFY-Core. Filtering was further performed for the presence of a phosphorylated tyrosine (motif) or phospho Ser/Thr within two residues of a tyrosine (lax) on the peptide. A > 2.5-fold cutoff was used for significant fold changes. PTMScan results are provided in Supplementary Data 3. The raw proteomics data are publicly available in the ProteomeXchange partner repository with an accession number of PXD051981.

### Phos-Tag gels
SuperSep 50 µM 7.5% 17 Well Phos-Tag gels (FujiFilm Wako Chemicals) were used to assess protein phosphorylation. After electrophoresis, Phos-Tag acrylamide gels were washed with transfer buffer containing 0.01% SDS and 1 mM EDTA for 10 min with gentle shaking, and then replaced with transfer buffer containing 0.01% SDS without EDTA for 20 min according to the manufacturer's protocol. Proteins were transferred to nitrocellulose membranes by wet transfer and analyzed by conventional immunoblotting.

### In vitro kinase assay
Recombinant KDR (357-KD-050/CF) was purchased from R&D Systems. Tax was transfected into 293T cells and lysates were immunoprecipitated with anti-Tax. An in vitro kinase assay was performed with the eluted immunoprecipitated Tax. Recombinant KDR and immunoprecipitated Tax were mixed (1:1 ratio) and incubated in 1× kinase buffer (200 mM HEPES, pH 7.4; 100 mM DTT,1 mM EGTA, 1 mM ATP, and 100 mM $MgCl_2$) for 2 h in a heated shaker at 37 °C. The reaction was terminated by adding loading buffer and phosphorylated proteins were detected by Phos-Tag gels.

### Flow cytometric staining of HAM/TSP PBMCs
PBMCs were labeled with fluorescent-conjugated antibodies (CD3-APC, CD4-Alexa Fluor 594, CD25-Brilliant Violet 421, and CD8-Brilliant Violet 711) as recommended by the eBioscience surface staining protocol and acquired on a FACS Symphony A3 flow cytometer. Data analysis was performed using FlowJo v10 software. The gating strategy for these experiments is in Supplementary Fig. 14.

### Statistical analysis
Data are expressed as the mean fold increase ± standard deviation relative to the control from a representative experiment performed three times in triplicate. Statistical analysis was performed using GraphPad Prism 10 and is indicated in the figure legends and supplemental figure legends.

### Reporting summary
Further information on research design is available in the Nature Portfolio Reporting Summary linked to this article.

## Data availability
All data are available in the main text or supplementary materials. The proteomics data are publicly available in the ProteomeXchange partner repository with an accession number of PXD051981. Source data are provided with this paper.

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

## Acknowledgements

The authors thank Dr. Lee Ratner for the provision of JET cells; Drs. Michiyuki Maeda and Masao Matsuoka for Tax- ATLL cell lines and Dr. Matthew Stokes and colleagues at Cell Signaling Technology for performing the PTMScan® Discovery Proteomics service. We are grateful to the HAM/TSP patients who contributed to this study. This work was supported by the National Institute of Allergy and Infectious Diseases (NIH R21 AI166335 to E.W.H.) and the NINDS intramural research program (S.J.). The Flow Cytometry Core (RRID:SCR_021134) services and instruments used in this project were funded, in part, by the Pennsylvania State University College of Medicine via the Office of the Vice Dean of Research and Graduate Students and the Pennsylvania Department of Health using Tobacco Settlement Funds (CURE). The content is solely the responsibility of the authors and does not necessarily represent the official views of the University or College of Medicine. The Pennsylvania Department of Health specifically disclaims responsibility for any analyses, interpretations, or conclusions.

## Author contributions

S.M., A.L., and E.W.H designed the experiments. S.M. performed most of the experiments with assistance from S.S. A.L. performed the kinome-wide shRNA screen and conducted initial experiments with KDR inhibitors. S.M., S.S., A.L., and E.W.H. analyzed the data. T.U. and J.F. provided JET reporter cells. N.N. and S.J. provided clinical samples. S.M. and E.W.H. wrote the manuscript. E.W.H. conceived and supervised the project and acquired funding for the project. All authors reviewed and approved the manuscript.

## Competing interests

The authors declare no competing interests.
