## [Peer Review File · Nature Communications]

The tyrosine kinase KDR is essential for the survival of HTLV-1-infected T cells by stabilizing the Tax oncoproteinREVIEWER COMMENTS

Reviewer #1 (Remarks to the Author):

In this study, Mohanty et al. use unbiased large-scale approaches such as upstream kinome-wide lentiviral screening and downstream phosphoproteomics, combined with targeted molecular and pharmaceutical blocking of a newly identified survival pathway in HTLV-1-transformed cells. In addition, the authors comprehensively study the molecular and cellular mechanisms through which VEGFR2 inhibition kills HTLV-1-transformed cells, culminating in a central role for Tax, the main viral protein linked to pathogenesis (both neuroinflammatory disease and leukemia/lymphoma). Mohanty et al. convincingly demonstrate a central role for autophagy in Tax degradation, documented by confocal microscopy. Last but not least, the authors succeeded in translating their in vitro findings to the ex vivo level, demonstrating clinically relevant elimination of CD4+CD25+ HTLV-1-infected T cells, upon treatment of HAM/TSP PBMCs with a KDR inhibitor.

In summary, the comprehensive data presented by Mohanty et al. provide an elegant stepwise validation of their research hypothesis and reveal a novel druggable pathway in HTLV-1-associated pathologies, of immediate clinical relevance to the millions of people living with HTLV-1 worldwide.

Major comment:

Despite the central role of KDR/VEGFR2 in the manuscript, the ligands of the VEGF family are not discussed at all. The authors mention VEGFR2 mutation in North-American ATL patients, do these mutations render the receptor inactive or do these mutations confer ligand-independent signaling? The authors rightfully state (line 88) "Vascular endothelial cell growth factors (VEGFs) critically regulate angiogenesis by binding to high-affinity receptor tyrosine kinases (RTKs), including VEGF receptors 1-3." However, a wealth of studies on angiogenesis and the role of VEGF in both HAM and ATL pathogenesis, as well as links to Tax have been published in HTLV infection and its associated pathologies (see references below). Can the authors link their compelling data on VEGFR2 to help resolve some of the conflicting data on VEGF/angiogenesis in the HTLV field?

Minor comments:

1. Ethical approval by the institutional Ethical Committee for the use of patient cells seems to be missing in the manuscript or at least a waiver, e.g. use of leftover diagnostic samples.
2. Can the authors explain why direct KDR interaction with Tax was not picked up in the several publications dedicated to the Tax interactome? Is the phenomenon cell type-specific?

References:

1: Kchour G, Makhoul NJ, Mahmoudi M, Kooshyar MM, Shirdel A, Rastin M, Rafatpanah H, Tarhini M, Zalloua PA, Hermine O, Farid R, Bazarbachi A. Zidovudine and interferon-alpha treatment induces a high response rate and reduces HTLV-1 proviral load and VEGF plasma levels in patients with adult T-cell leukemia from North East Iran. *Leuk Lymphoma*. 2007 Feb;48(2):330-6. doi: 10.1080/10428190601071717. PMID: 17325893.

- 2: El-Sabban ME, Merhi RA, Haidar HA, Arnulf B, Khoury H, Basbous J, Nijmeh J, de Thé H, Hermine O, Bazarbachi A. Human T-cell lymphotropic virus type 1-transformed cells induce angiogenesis and establish functional gap junctions with endothelial cells. *Blood*. 2002 May 1;99(9):3383-9. doi: 10.1182/blood.v99.9.3383. PMID: 11964307.
- 3: Saito M, Usuku K, Nobuhara Y, Matsumoto W, Kodama D, Sabouri AH, Izumo S, Arimura K, Osame M. Serum concentration and genetic polymorphism in the 5'-untranslated region of VEGF is not associated with susceptibility to HTLV-I associated myelopathy/tropical spastic paraparesis (HAM/TSP) in HTLV-I infected individuals. *J Neurol Sci*. 2004 Apr 15;219(1-2):157-61. doi: 10.1016/j.jns.2004.01.009. PMID: 15050452.
- 4: Mitra-Kaushik S, Harding J, Hess J, Schreiber R, Ratner L. Enhanced tumorigenesis in HTLV-1 tax-transgenic mice deficient in interferon-gamma. *Blood*. 2004 Nov 15;104(10):3305-11. doi: 10.1182/blood-2004-01-0266. Epub 2004 Aug 3. PMID: 15292059.
- 5: Ghez D, Lepelletier Y, Lambert S, Fourneau JM, Blot V, Janvier S, Arnulf B, van Endert PM, Heveker N, Pique C, Hermine O. Neuropilin-1 is involved in human T-cell lymphotropic virus type 1 entry. *J Virol*. 2006 Jul;80(14):6844-54. doi: 10.1128/JVI.02719-05. PMID: 16809290; PMCID: PMC1489069.
- 6: Kchour G, Tarhini M, Sharifi N, Farid R, Khooei AR, Shirdel A, Afshari JT, Sadeghian A, Otrrock Z, Hermine O, El-Sabban M, Bazarbachi A. Increased microvessel density in involved organs from patients with HTLV-I associated adult T cell leukemia lymphoma. *Leuk Lymphoma*. 2008 Feb;49(2):265-70. doi: 10.1080/10428190701760060. PMID: 18231912.
- 7: Zhao T, Yasunaga J, Satou Y, Nakao M, Takahashi M, Fujii M, Matsuoka M. Human T-cell leukemia virus type 1 bZIP factor selectively suppresses the classical pathway of NF-kappaB. *Blood*. 2009 Mar 19;113(12):2755-64. doi: 10.1182/blood-2008-06-161729. Epub 2008 Dec 8. PMID: 19064727.
- 8: Lambert S, Bouttier M, Vassy R, Seigneuret M, Petrow-Sadowski C, Janvier S, Heveker N, Ruscetti FW, Perret G, Jones KS, Pique C. HTLV-1 uses HSPG and neuropilin-1 for entry by molecular mimicry of VEGF165. *Blood*. 2009 May 21;113(21):5176-85. doi: 10.1182/blood-2008-04-150342. Epub 2009 Mar 6. PMID: 19270265; PMCID: PMC2686187.
- 9: Watters KM, Dean J, Gautier V, Hall WW, Sheehy N. Tax 1-independent induction of vascular endothelial growth factor in adult T-cell leukemia caused by human T-cell leukemia virus type 1. *J Virol*. 2010 May;84(10):5222-8. doi:

10.1128/JVI.02166-09. Epub 2010 Mar 17. PMID: 20237090; PMCID: PMC2863836.

10: Watters KM, Dean J, Hasegawa H, Sawa H, Hall W, Sheehy N. Cytokine and growth factor expression by HTLV-1 Lck-tax transgenic cells in SCID mice. *AIDS Res Hum Retroviruses*. 2010 May;26(5):593-603. doi: 10.1089/aid.2009.0212. PMID: 20438380.

11: Meireles AL, Hallack Neto AE, Costa Rde O, Pereira J. Increased levels of circulating endothelial progenitor cells in human T-cell lymphotropic virus type I carriers. *Arch Med Res*. 2011 Jan;42(1):34-7. doi: 10.1016/j.arcmed.2011.01.002. PMID: 21376260.

12: Zong Y, Kamoi K, Kurozumi-Karube H, Zhang J, Yang M, Ohno-Matsui K. Safety of intraocular anti-VEGF antibody treatment under *in vitro* HTLV-1 infection. *Front Immunol*. 2023 Jan 25;13:1089286. doi: 10.3389/fimmu.2022.1089286. PMID: 36761168; PMCID: PMC9905742.

13: Freitas NL, Gomes YCP, Souza FDS, Torres RC, Echevarria-Lima J, Leite ACCB, Lima MASD, Araújo AQC, Silva MTT, Espíndola OM. Lessons from the Cerebrospinal Fluid Analysis of HTLV-1-Infected Individuals: Biomarkers of Inflammation for HAM/TSP Development. *Viruses*. 2022 Sep 29;14(10):2146. doi: 10.3390/v14102146. PMID: 36298702; PMCID: PMC9609689.

14: Shimizu Y, Yamanashi H, Miyata J, Takada M, Noguchi Y, Honda Y, Nonaka F, Nakamichi S, Nagata Y, Maeda T. VEGF Polymorphism rs3025039 and Human T-Cell Leukemia Virus 1 (HTLV-1) Infection among Older Japanese Individuals: A Cross-Sectional Study. *Bioengineering (Basel)*. 2022 Oct 6;9(10):527. doi: 10.3390/bioengineering9100527. PMID: 36290496; PMCID: PMC9598135.

Reviewer #2 (Remarks to the Author):

Major Comments

HTLV-1Tax is essential for replication of the virus itself and has a significant impact on the function of infected target cells. Delineation the function of Tax is essential information for understanding the mechanism of dysfunction of infected cells and the viral pathogenesis based on it. However, its function remains poorly understood.

In this study, the authors searched for kinases that interact with Tax by a kinome-wide shRNA screen and discovered a novel interaction between KDR and Tax.

The authors report a series of experiments to clarify the biological significance of this interaction, including.

1. the critical role of KDR in HTLV-1 transformed cells
2. the protective mechanism of Tax mediated by KDR.
3. involvement of KDR in HTLV-1 infection by protecting p19/Gag protein expression.
4. induction of KDR expression by Tax and tyrosine phosphorylation of Tax via KDR
5. interaction and co-localization of Tax and KDR in the Golgi apparatus
6. activation of IKK and regulation of autophagosome biogenesis by this interaction

In general, the experimental design was well considered, and the interpretation of the results is reasonable. The results obtained may provide valuable new information for understanding the mechanisms of Tax-mediated activation of various signaling pathways and anti-apoptotic functions. These findings can be evaluated as providing new perspectives on the mechanisms of viral replication and infection, as well as on feasible therapeutic strategies for HTLV-1-induced inflammatory diseases such as HAM/TSP and HTLV-1 uveitis, and therapy-resistant T-cell lymphoma (ATL).

Minor Comments

1. In Fig. 7 F. the figure is too small to read the specific description of the genes. This should be presented in a larger figure.

2. Fig.9. and ex vivo analysis of KDR inhibitors.

In Fig 9 A, authors showed effects of SU1498 on the PBMC from HAM patients and healthy controls. Authors described that HAM PBMC were cultured for 5 days before analysis. It is well known that HAM PBMCs will spontaneously proliferate in the culture without IL-2. However, it is not expected for the cases of healthy control PBMCs. Thus, additional description of the condition as to the control PBMCs is expected.

We have addressed all the comments made by the reviewers as summarized below in the point-by-point responses.

Reviewer 1:

1. Despite the central role of KDR/VEGFR2 in the manuscript, the ligands of the VEGF family are not discussed at all. The authors mention VEGFR2 mutation in North-American ATL patients, do these mutations render the receptor inactive or do these mutations confer ligand-independent signaling? The authors rightfully state (line 88) “Vascular endothelial cell growth factors (VEGFs) critically regulate angiogenesis by binding to high-affinity receptor tyrosine kinases (RTKs), including VEGF receptors 1-3.” However, a wealth of studies on angiogenesis and the role of VEGF in both HAM and ATL pathogenesis, as well as links to Tax have been published in HTLV infection and its associated pathologies (see references below). Can the authors link their compelling data on VEGFR2 to help resolve some of the conflicting data on VEGF/angiogenesis in the HTLV field?

We thank the reviewer for these important comments on the VEGF family in HTLV-1-associated diseases. In the revised manuscript, we have included a discussion (see lines 411-423) on the roles of VEGF in ATLL and HAM/TSP and how they could relate to our findings. We have also cited key references mentioned by the reviewer. Based on our data, we don't believe that VEGF will play a role in Tax stabilization since Tax sequesters KDR inside the cell in a perinuclear area where it is unable to interact with any ligands. The VEGFR2 mutations in North American patients have not been characterized but we speculate that the mutations may be gain-of-function that would confer ligand-independent signaling.

2. Ethical approval by the institutional Ethical Committee for the use of patient cells seems to be missing in the manuscript or at least a waiver, e.g. use of leftover diagnostic samples.

We thank the reviewer for noticing the omission of this section in the manuscript. The revised manuscript has an Ethics Statement in the Methods section that describes the provision of the HAM/TSP blood samples.

3. Can the authors explain why direct KDR interaction with Tax was not picked up in the several publications dedicated to the Tax interactome? Is the phenomenon cell type-specific?

The reviewer raises a good point. We have also performed a mass spectrometry screen to identify Tax interacting proteins (Gao et al. 2013 *J. Virol.* 87: 13640-54) but did not identify KDR in this screen. We speculate that KDR was not previously identified as a Tax interacting protein because it is mainly associated with cell membranes and KDR is barely detectable unless membrane fractions are isolated.

Reviewer 2:

1. In Fig. 7 F. the figure is too small to read the specific description of the genes. This should be presented in a larger figure.

Thank you for the comment. Fig. 7F has now been enlarged to make it easier for readers to read the protein names in the heat map. To make space for the enlarged figure, the previous Fig. 7B and 7C have been moved to Supplementary Fig. 6.

2. In Fig 9 A, authors showed effects of SU1498 on the PBMC from HAM patients and healthy controls. Authors described that HAM PBMC were cultured for 5 days before analysis. It is well known that HAM PBMCs will spontaneously proliferate in the culture without IL-2. However, it is not expected for the cases of healthy control PBMCs. Thus, additional description of the condition as to the control PBMCs is expected.

We thank the reviewer for the comment. IL-2 was not added to any of our PBMC cell cultures. Frozen vials of HAM/TSP PBMCs were thawed and subjected to experiments after five days of culture in order to induce robust Tax expression. As mentioned by the reviewer, these cells spontaneously proliferate without IL-2. Frozen vials of control PBMCs were thawed and subjected to experiments after two days of culture since these cells will not grow without supplementation of IL-2. These details have now been included in the Methods section.

REVIEWERS' COMMENTS

Reviewer #1 (Remarks to the Author):

The authors have addressed all pertinent questions and have adapted the manuscript accordingly. I agree with publication in its current form.

Reviewer #2 (Remarks to the Author):

The revised manuscript is considered to adequately address the reviewers' comments throughout and appears to have improved significantly.

I believe that the information presented in the paper provides new insights into the interaction between the viral gene product and HTLV-1-infected cells.

The reviewers were both in agreement that we addressed the previous concerns and the manuscript is now acceptable for publication.

Reviewer 1:

1. The authors have addressed all pertinent questions and have adapted the manuscript accordingly. I agree with publication in its current form.

We are glad we have addressed all pertinent questions and thank the reviewer for recommending publication in its current form.

Reviewer 2:

1. The revised manuscript is considered to adequately address the reviewers' comments throughout and appears to have improved significantly. I believe that the information presented in the paper provides new insights into the interaction between the viral gene product and HTLV-1-infected cells.

We thank the reviewer for these comments.